# Objective identification of pressure wave events from networks of 1-Hz, high-precision sensors

Luke R. Allen[1], Sandra E. Yuter[1,2], Matthew A. Miller[2], and Laura M. Tomkins[1]

[1]Center for Geospatial Analytics, North Carolina State University, Raleigh, NC, 27695, USA
[2]Department of Marine, Earth, and Atmospheric Sciences, North Carolina State University, Raleigh, NC, 27695, USA

**Correspondence:** Luke R. Allen (lrallen3@ncsu.edu)

**Abstract.** Mesoscale pressure waves, including atmospheric gravity waves, outflow and frontal passages, and wake lows, are outputs of and can potentially modify clouds and precipitation. The vertical motions associated with these waves can modify the temperature and relative humidity of air parcels and thus yield potentially irreversible changes to the cloud and precipitation content of those parcels. A wavelet-based method for identifying and tracking these types of wave signals in time series data from networks of low-cost, high-precision (0.8-Pa noise floor, 1-Hz recording frequency) pressure sensors is demonstrated. Strong wavelet signals are identified using a wave period-dependent (i.e., frequency-dependent) threshold, then those signals are extracted by inverting the wavelet transform. Wave periods between 1 minute and 120 minutes were analyzed, a range which could capture acoustic, acoustic-gravity, and gravity wave modes. After extracting the signals from a network of pressure sensors, the cross-correlation function is used to estimate the time difference between the wave passage at each pressure sensor. From those time differences, the wave phase velocity vector is calculated using a least-squares fit. If the fitting error is sufficiently small (thresholds of RMSE < 90 s and NRMSE < 0.1 were used), then a wave event is considered robust and trackable. We present examples of tracked wave events, including a Lamb wave caused by the Hunga-Tonga volcanic eruption in January 2020, a gravity wave train, an outflow boundary passage, a frontal passage, and a cold front passage. The data and processing techniques presented here can have research applications in wave climatology and testing associations between waves and atmospheric phenomena.

## 1 Introduction

Gravity waves (i.e., buoyancy waves), which result from vertical perturbations of stably stratified fluid, are ubiquitous in the atmosphere and represent an important distributor of energy through the atmosphere (Fritts and Alexander, 2003; Nappo, 2013). The initial perturbations which generate gravity waves can have several sources, including but not limited to forced flow over topography, deep convection, shear instability, adjustment of unbalanced flow, and nonlinear interaction between waves (Fritts and Alexander, 2003). Within the troposphere, the vertical motions associated with gravity waves have been shown to influence cloud and precipitation processes. If parcels then reach saturation and produce precipitation which falls out, the changes to the parcel can be irreversible. Several studies have investigated the effects of gravity waves on marine stratocumulus. Allen et al. (2013) and Connolly et al. (2013) related gravity waves to changes in drizzle production within

marine stratocumulus; specifically, enhanced condensation and collision-coalescence to form drizzle drops appeared to occur in the updrafts associated with gravity waves. Evidence also suggests a link between gravity waves and the rapid erosion of marine stratocumulus cloud decks (Yuter et al., 2018), perhaps because evaporation due to entrainment is enhanced as marine stratocumulus clouds are lifted by gravity waves (Connolly et al., 2013). Fovell et al. (2006) identified gravity waves as a potential trigger mechanism for deep moist convective cells. Case studies have linked gravity waves to snow bands, i.e., linear

mesoscale enhancements in snowfall rate within winter storms (Bosart et al., 1998; Zhang et al., 2001; Gaffin et al., 2003), but it is unclear how often snow bands are associated with gravity waves.

To illustrate how the vertical motions associated with gravity waves could influence cloud microphysical properties, we use Fig. 1, which shows relative humidity with respect to ice ($RH_{ice}$) and with respect to liquid water ($RH_{water}$) as a function of temperature and water vapor mixing ratio for a standard atmosphere, as well as the temperature difference between 100%

$RH_{ice}$ and 100% $RH_{water}$ for each water vapor mixing ratio. For temperatures below 0°C, $RH_{ice} > RH_{water}$. A lifted parcel would be cooled at a constant water vapor mixing ratio (move upward in Fig. 1) until it intersects the 100% $RH_{ice}$ contour wherein vapor deposition would reduce the water vapor (further lifting would move upward and to the left in Fig. 1). If the parcel cools sufficiently to also intersect the 100% RHwater contour, supercooled water droplets would form in the parcel and riming would likely occur further depleting the available water vapor in the parcel. An up and down motion of an air parcel in

a gravity wave that only crosses the 100% $RH_{ice}$ contour yields ice mass changes that are reversible, i.e. the ice mass added by vapor deposition in the upward motion will be lost to sublimation in the downward motion.

In contrast, if the lifting of the parcel within a gravity wave starts at $RH_{ice} \geq 100\%$ and intersects the 100% $RH_{water}$ contour, the ice mass added by riming when ice particles collide with supercooled droplets is not reversible (i.e., there is no unriming process). For example, if an ice saturated parcel containing ice crystals (e.g., at -8.0°C and 2.96 g kg$^{-1}$ water vapor

mixing ratio) is lifted and cooled to -9.7°C in a gravity wave (which requires 190 m of lift assuming a 9°C km$^{-1}$ parcel lapse rate), it would become supersaturated with respect to water as well as with respect to ice. Water droplets would form in the parcel and the ice crystals could potentially become rimed. If the parcel remains ice saturated when it descends in the gravity wave, the additional ice mass from the riming on the ice particle would not be removed. Ice mass added by riming can only be removed by sublimation in conditions where the parcel is subsaturated with respect to ice.

One way of distinguishing gravity waves from other wave phenomena such as Kelvin-Helmholtz waves is that gravity waves produce a surface pressure signal (Nappo, 2013, Sect. 8.2), given that the stable layer in which they occur is adjacent or nearly adjacent to the surface. That said, several different phenomena can produce surface pressure disturbances on similar spatiotemporal scales to gravity waves, including but not limited to outflow boundary passages, convective wake lows (Johnson and Hamilton, 1988), release of conditional symmetric instability (Gray et al., 2011), and Lamb waves generated by, e.g.,

distant volcanic eruptions (Matoza et al., 2022), large bolide impacts (ReVelle, 2008), and thermonuclear explosions (Pierce and Posey, 1971). Acoustic and acoustic-gravity waves can also produce pressure signals at shorter time scales. Such waves would include infrasound waves, which can be associated with, e.g., convective storms and strong flow over mountains (Coffer and Parker, 2022; Bedard, 1978).

Time series of surface pressure data have been analyzed to identify tropospheric wave events in previous studies (Kjelaas et al., 1974; Christie et al., 1978; Einaudi et al., 1989; Grivet-Talocia and Einaudi, 1998; Grivet-Talocia et al., 1999; Koch and Saleeby, 2001; de Groot-Hedlin et al., 2014). Kjelaas et al. (1974) and Christie et al. (1978) presented case studies of gravity wave events selected manually from time series pressure data. Grivet-Talocia and Einaudi (1998) and Grivet-Talocia et al. (1999) recorded data at $1/120$ Hz (i.e., every 2 min) to identify wave periods longer than 30 min. Einaudi et al. (1989) used a network of microbarographs recording at 0.1 Hz (i.e., every 10 sec) placed within roughly 100 m of each other, which constrained the characteristics of disturbances which could be tracked through their network to waves with speeds up to $50\,\mathrm{m\,s^{-1}}$, and wave periods of 1-20 min. Koch and Saleeby (2001) used operational Automatic Surface Observing Systems (ASOS) data recorded at $1/300$ Hz (i.e., every 5 min) which resolved wavelengths $\geq 150\,\mathrm{km}$. While strong pressure disturbances including outflows and wake lows can be detected by ASOS pressure sensors logging data at 1 min intervals, the large spatial separation between operational weather stations, which are mostly located at airports, precludes determination of the associated wave speed and direction for mesoscale disturbances. de Groot-Hedlin et al. (2014) used 337 barometers deployed with the USArray Transportable Array, recording at 1 Hz (i.e., every 1 sec) frequency, to detect and track high amplitude (roughly 3 hPa peak to trough) pressure waves associated with convective storms in the southern United States. The USArray Transportable Array barometers were spaced roughly 70 km apart, which might also preclude tracking of localized disturbances.

There is a scarcity of data for detecting and tracking pressure disturbances, including gravity waves, on the meso-$\beta$-scale or meso-$\gamma$-scale (ranging from 2 km to 200 km). Pressure disturbances on those scales may be relevant to phenomena such as snow bands (e.g. McMurdie et al., 2022), trade wind cumulus (e.g. Seifert and Heus, 2013), and bow echoes (e.g. Adams-Selin and Johnson, 2010), which are active areas of research. We developed a measurement and analysis technique which will allow for questions regarding gravity waves on those scales to be addressed. To what degree and in what conditions information on gravity wave occurrence would be valuable in operational weather settings is yet to be determined.

This paper presents data from networks of internet appliance low cost, high precision air pressure sensors (i.e., microbarographs) and a methodology for objectively identifying mesoscale wave events and wave speed and direction. The methodology is intended to be used for post-processing in research applications, rather than for real time or near real time detection of wave events. Similar low cost sensor networks have been used for detection of seismic (Anthony et al., 2018) and for detecting infrasound waves to monitor fan rotation speeds in nuclear reactor cooling towers (Eaton et al., 2022). The former network covered the area of Oklahoma, and the latter used networks covering the area of a single nuclear reactor. Our networks of pressure sensors are on the scale of a medium to large sized city or metropolitan area.

Section 2 of this paper describes the pressure sensors used in this study and the data they provide. Section 3 describes the methodology for objectively identifying pressure waves from the pressure time traces. Section 4 provides five examples of events captured by the wave identification method. Finally, a summary and avenues for future work are discussed in Sect. 5.

## 2 Data

### 2.1 Networks of pressure sensors

Pressure sensors were placed in three separate networks: New York City metro area and Long Island, NY, Raleigh, NC, and Toronto, ON, Canada (Fig. 2). Each pressure sensor was either a Bosch BME280 (Bosch, 2022) or a Bosch BMP388 (Bosch, 2020) Adafruit breakout board connected to a Raspberry Pi Zero W single-board computer used to log the data. BME280 sensors measure pressure, temperature, and humidity. BMP388 sensors measure pressure and temperature. Each pressure sensor costs roughly 50-75 US Dollars, subject to changes in the cost of Raspberry Pi Zero W units. For context to another low cost network concept, the Raspberry Shake 4D seismographs cost "a few hundred dollars" per unit (Anthony et al., 2018). The combined sensor and communications package is about the size of a deck of cards. Sensors are connected to the internet and sync their data to a server at North Carolina State University. Initial testing of the sensors outdoors on patios, in sheltered locations such as garages, and indoors revealed pressure waves were well resolved in all locations and it was best practice to place the sensors indoors to minimize wind contamination in pressure measurements. When active, each sensor records pressure at 1-second intervals with a roughly 0.8 Pa noise floor depending on ambient conditions. The sensors synchronize to network time upon startup. The wave extraction method only depends on relative pressure variations and is not sensitive to absolute or relative calibration.

To examine the properties of gravity waves which are detectable by these pressure sensors, we consider an internal gravity wave occurring in an environment with constant background wind $u_0$ The relationship between the pressure perturbation $p'$ and the horizontal velocity perturbation $u'$ associated with the wave is described by Nappo (2013):

$$p' = u'\rho_0(c - u_0) \tag{1}$$

where $\rho_0$ is the environmental air density and $c$ is the phase speed of the gravity wave. From Eq. 1, the maximum pressure perturbation $p'_{max}$ can therefore be related to the maximum horizontal velocity perturbation $u'_{max}$ by:

$$p'_{max} = u'_{max}\rho_0(|c - u_0|) \tag{2}$$

Figure 3 shows $p'_{max}$ values according to Eq. 2 at an air density of $1.225 \ \mathrm{kg \ m^{-3}}$ (standard air density at sea level; American Meteorological Society, 2022) for $u'_{max}$ and $|c - u_0|$ values up to $15 \ \mathrm{m \ s^{-1}}$. In order to reliably detect a wave signal, the amplitude likely needs to be substantially larger than the noise floor.

### 2.2 Operational Weather Observations

For context, we compare extracted wave signals with available operational weather observations.

We use Automated Surface Observing Systems (ASOS; NOAA National Centers for Environmental Information, 2021a) data including surface temperature, dew point, wind speed and direction, and additional pressure measurements coincident with

wave events. ASOS data are recorded each minute. For wave events detected in New York and Long Island, we examined ASOS data from John F. Kennedy International Airport (KJFK) and Long Island MacArthur Airport (KISP). For wave events detected in Toronto, we examined ASOS data from Buffalo Niagara International Airport (KBUF) and Niagara Falls International Airport (KIAG).

For one example case in Sect. 4.2, we show upper-air radiosonde data from a weather balloon launched in Buffalo, NY. We obtained data from the Integrated Global Radiosonde Archive (IGRA; NOAA National Centers for Environmental Information, 2021b) and interpolated to a constant 100-meter resolution. The data include measurements of temperature, dew point, and winds, from which we calculated wet bulb temperature, frost point, and saturation equivalent potential temperature ($\theta_e^*$). Because radiosondes are typically launched every 12 hours at a limited number of locations, representative radiosonde data are not available for every case.

We use horizontal maps of data from the U.S. National Weather Service (NWS) WSR-88D radars (NOAA National Weather Service Radar Operations Center, 1991) to show storm features occurring coincident with wave events in Sect. 4.3 and Sect. 4.5. Radar reflectivities are processed following Tomkins et al. (2022) to indicate regions with mixed precipitation in the scan, by inferring that points with reflectivity above 20 dBZ and dual-polarization correlation-coefficient below 0.97 have mixed precipitation. In maps of radar reflectivity, those regions with mixed precipitation are then shown in greyscale. Doppler velocity waves are extracted from radial velocity data following Miller et al. (2022), by calculating the difference in radial velocity from successive scans, converting those differences to a binary (positive/negative) field, and filtering out small objects in that binary field.

## 3 Methods

The methods outlined here for identifying wave events in the pressure time traces are adapted from the techniques used by Grivet-Talocia and Einaudi (1998) and Grivet-Talocia et al. (1999). The method uses wavelet transforms to identify wave events in time-wave period (or, equivalently, time-frequency) space. Wavelet transforms are preferable to Fourier transforms for the purpose of identifying transient waves which are localized in time (Torrence and Compo, 1998). To illustrate the step-by-step procedure, an example corresponding to an gravity wave event on 23 February 2023 in the Toronto pressure network is described in detail.

### 3.1 Identifying wave events in a single sensor

The full pressure time series for a gravity wave event on 23 February 2023 captured by sensor 25 in Toronto is shown in Fig. 4a. As an initial pre-processing step, 10-second samples of pressure (i.e., averages of 10 pressure measurements) are used to smooth out noise and pressure perturbations due to high frequency turbulent eddies in the data (Fig. 4b). Hereafter, time series labeled as total pressure are the 10-second sub samples of the original pressure measurements.

A wavelet transform $W$ of a finite energy signal $f(t)$ (a pressure time series in this study) can be defined as (Grivet-Talocia and Einaudi, 1998, their Eq. 1):

$$W(b,a) = \frac{1}{|a|} \int\limits_{-\infty}^{\infty} f(t)\psi^*(\frac{t-b}{a})dt \tag{3}$$

where $a$ is the scale (related to the wave period), and $b$ shifts the wavelet in time ($t$). $\psi^*$ represents the *mother wavelet*. An analytic Morse wavelet was used (e.g. Olhede and Walden, 2002; Lilly and Olhede, 2012) via the *cwt* function within Matlab (Lilly, 2021). In this study, $W(b,a)$ will always refer to the wavelet transform of a pressure time series. The resulting wavelet transform is an array of complex values in time-scale space. The absolute value of the wavelet transform $|W(b,a)|$ can be considered the wavelet *power* at a given time and scale. Figure 4c shows the wavelet power associated with the wave event at sensor 25 on 23 February 2023. In this study, wave periods between 1 minute and 120 minutes were analyzed corresponding to expected periods for mesoscale disturbances.

To objectively identify wave event centers according to wavelet power, a scale-dependent (i.e., wave period-dependent) threshold function $A(a)$ is defined as the mean wavelet power across all available data for the sensor network by scale, multiplied by a constant $K$:

$$A(a) = K \langle |W(b,a)| \rangle_b \tag{4}$$

A scale-dependent threshold $K$ is necessary because the 'background' wavelet power for a pressure time series generally increases with scale (e.g., Canavero and Einaudi, 1987). Grivet-Talocia and Einaudi (1998) and Grivet-Talocia et al. (1999) used 2 as an appropriate value for $K$. Lower values of $K$ lead to more wave events being detected, which can include potential artifacts in the pressure time trace. For the present study, a $K$ value of 10 was used to ensure that only the strongest wave signals were identified (solid contour in Fig. 4d). This threshold can be adjusted, and different applications may warrant different values of $K$. The mean wavelet power as a function of wave period is shown for each regional sensor network and for all networks combined in Fig. 5. Event centers were identified as local maxima in wavelet amplitude which exceed $A(a)$, which are located at $(b_{max}, a_{max})$. In Fig. 4, an event center is located within the solid contour. From the identified event centers, the first iterations of event regions ($\Omega'$) were identified in time-scale space as connected regions where the wavelet power exceeds $\frac{K}{2} \langle |W(b,a)| \rangle_b$, i.e., half of the event center threshold. In Fig.4, $\Omega'$ is represented by the region within a dashed contour which contains a solid contour.

The watershed transform (Meyer, 1994) was used to refine $\Omega'$. Watersheds (i.e., catchment basins) were identified in the negative wavelet power array $-|W(b,a)|$. Any watersheds within $\Omega'$ whose period range was entirely outside the period range of the watershed containing the event center were removed from the event region $\Omega'$. This step was included to correct cases where multiple "peaks" in wavelet power were present within $\Omega'$ at different wave periods, with a "valley" in wavelet power in between where the wavelet power still exceeded $\frac{K}{2} \langle |W(b,a)| \rangle_b$, which likely represented distinct wave modes and should be considered separate wave events.

$\Omega'$ was extended to define the final event region $\Omega$ for each event, first by taking the bounding box of $\Omega'$, then by extending the bounding box along the time axis in both directions until it reaches a local minimum in the wavelet amplitude to ensure that

the entire signal of interest is contained in the event region. This could result in overlapping event regions. Figure 4d shows the wavelet power normalized by the mean wavelet power by scale ($|W|$ / $\langle|W|\rangle_b$) for the 23 February 2023 example in sensor 25, with the outline of the event region overlaid with the magenta box.

After defining the event region $\Omega$, the wave event trace could then be extracted (i.e., reconstructed) by inverting the wavelet transform function over the event region $\Omega$. Figure 4e shows the extracted wave event trace for the 23 February 2023 wave event in sensor 25. As in Grivet-Talocia and Einaudi (1998), wave events were identified and extracted one at a time, with the extracted wave event subtracted from the pressure trace and the wavelet transform recalculated at each iteration, until the absolute maximum of $|W|$ / $\langle|W|\rangle_b$ was less than $K$ (i.e., until no more events are left to be found in the pressure time series).

We tested the method of detecting wave events in a single sensor using synthetic pressure data. The synthetic time series of pressure is created by an initial constant pressure value (which is randomly chosen from a normal distribution with mean 1000 hPa and standard deviation 2 hPa). We then added normally distributed random noise centered on 0 with standard deviation equal to the noise floor (0.008 hPa). Finally, we added 105 pre-defined wave events, which consist of sine waves of period ranging from 2 min to 120 min and maximum amplitude ranging from $\pm0.01$ hPa to $\pm1$ hPa. Each set of sine waves lasts

for 2 hours, with a 12 minute ramp-up and ramp-down period at the start and end of those 2 hours in which the amplitude increases and decreases linearly, respectively. One of these synthetic wave events is shown in Fig. 6a. Using the $\langle|W(b,a)|\rangle_b$ values shown in Fig. 5 and a $K$ value of 10, 52 of the 105 synthetic wave events were detected, with no false positive event detections (Fig. 6c). The weakest detected synthetic wave event had an amplitude of $\pm0.0464$ hPa (or 0.0928 from peak to trough) and a wave period of 2 min (shown in Fig. 6a). Lower values of $K$ do lead to more wave events being detected, with

few false positives. The $K = 2$ used by Grivet-Talocia and Einaudi (1998) results in 86 of the 105 wave events being detected with only one false positive. However, this exercise likely fails to capture the full extent of noise and the interference of many signals present in real pressure data. Lower values of $K$ can result in more weak pressure wave events being detected, which may be "real" at a single sensor location, but these may then erroneously paired with other weak pressure wave events at other sensor locations using the methods described in the proceeding sections. Therefore, we will use $K = 10$ to detect and track

pressure wave events across the networks of sensors.

## 3.2    Matching corresponding wave events between multiple sensors

Once wave events were identified for each sensor individually, the following steps were taken to identify coherent wave events across multiple sensors. For this purpose, the terms *primary sensor* (denoted by i) and *secondary sensor* (denoted by j) will be used to describe a pair of sensors for which events are identified and paired together.

For each event in the primary sensor pressure trace, events in the secondary sensor pressure trace that occurred within 2 hours of that primary sensor event (i.e., with a gap between the end of the event in one sensor and the start of the event in the other sensor not exceeding 2 hours) were considered "candidate" events to match with the primary sensor event. This 2 hour threshold is subjective, and it affects the range of speeds of wave events which can be detected. The threshold can be altered depending on the distances between pressure sensors and the desired application. For example, the largest distance between

any 2 sensors in the 3 networks is that between sensors 012 and 027 (136 km apart, Fig. 2). A wave feature propagating at

$18.9 \, \mathrm{m \, s^{-1}}$ would take 2 hours to propagate that distance. However, wave features propagating at an angle could be slower and propagate over both sensors within 2 hours, and as long as pressure sensors in between those two most distant sensors capture the event, the following processing technique will allow those sensors to "bridge the gap" even if the event takes longer than 2 hours to propagate across the distance between those sensors. Thus, $18.9 \, \mathrm{m \, s^{-1}}$ is a conservative estimate of the minimum

phase speed required for this methodology to track a wave event. Candidate matching events in the secondary sensor trace had to have a center period which was within the primary sensor event period range, and vice versa. Then, for each candidate matching event in the secondary sensor trace, the event waveforms are reconstructed by inverting the wavelet transform over the event region for both sensors. Figure 7 shows the extracted waveform for the 23 February 2023 event in sensor 25 and the same wave passage in sensors 04, 23, 24, and 34. The time lag estimate for the wave passage between sensors is $\Delta t_{opt}$, the

time lag which maximizes the cross-correlation function $C_{ij}(\Delta t)$:

$$C_{ij}(\Delta t) = \frac{1}{||p_i|| \, ||p_j||} \int p_i(t) p_j(t + \Delta t) dt \tag{5}$$

where $p_i(t)$ and $p_j(t)$ are the extracted waveforms for the events in the primary and secondary sensor, respectively. The black lines and subfigure titles in Fig. 7 show the optimal shift in the extracted waveforms for sensors 04, 23, 24, and 34 to maximize $C_{ij}$ to sensor 25 for the 23 February 2023 example. The match to the primary sensor event is the candidate event with the

highest maximized cross-correlation to the primary sensor event. If the maximized cross-correlation exceeded 0.65, and the same pair of matched events results from switching the primary and secondary sensors (i.e., the event is matched two-ways), the event from sensor i and the event from sensor j are paired together. Switching the primary and secondary sensors is necessary to avoid instances where multiple events in one sensor are matched with the same event in another sensor. This can occur, for example, when a set of waves manifests as one event in one sensor and multiple (separate) events in another sensor.

The process of matching events between sensors described above was repeated for each possible combination (within a sensor network) of primary and secondary sensors in order to obtain the full set of lag times between each pair of sensors which captured each event. In other words, $N^2$ pairs of sensors, order-dependent, were analyzed, where $N$ is the number of sensors in the network with data at a given time. Then, each event in each sensor was assigned an ID based on which other sensors had a matching event in order to track events across 3 or more sensors. This process required iterating through each

sensor in a network. Each event in the first sensor was assigned a new (i.e., arbitrary) ID. For each subsequent sensor $s_c$, events with no two-way matches in any prior sensor were also given new IDs. If there were two-way matches with an event in one or more prior sensor(s), the event in the sensor $s_c$ would share the ID assigned to the matched event in the prior sensor. If there were multiple prior sensors with matched events, and those events had different IDs, the ID associated with the higher maximized cross-correlation between the event traces was assigned to the event in sensor $s_c$. If this process results in multiple

events in sensor $s_c$ sharing the same event ID $D$, the event in sensor $s_c$ associated with the highest maximized cross-correlation with any one prior sensor for an event with event ID $D$ is assigned event ID $D$, and the previously outlined steps are repeated for the other event(s) in sensor $s_c$, with event ID $D$ and associated sensors excluded.

The result of this process is a set of events with associated ID numbers for every sensor in the network. For a single sensor, each event has a unique ID. For each ID number that appeared in at least three sensors, the wave phase velocity vector was

calculated using the set of lag times between each pair of sensors which captured the event.

### 3.3    Estimating wave phase velocity vector

Once sets of matched events were identified, the wave propagation velocities (two-dimensional vectors) could be estimated for events which occurred in three or more sensors. It is hypothesized for each wave event that a plane wave crosses the sensor network with slowness vector $s = (s_x, \ s_y)$, where $s_x$ and $s_y$ are the inverses of the x- and y-components of the wave

propagation vector (in s m$^{-1}$), respectively. $s$ can be solved for from the following equation (Del Pezzo and Giudicepietro, 2002):

$$t = s \cdot \Delta \mathbf{x} \tag{6}$$

where $t$ is the column vector of the $\Delta t_{opt}$ values for each possible pair of sensors which captured the event, and $\Delta x$ is the two-column matrix of the x- and y-components of the distance vector between each pair of sensors which captured the

event. $t$ and $\Delta \mathbf{x}$ each have $N_s(N_s - 1)/2$ rows, where $N_s$ is the number of sensors which captured the event. Equation 6 can be considered an overdetermined system of $N_s(N_s - 1)/2$ linear equations, as long as $N_s \geq 3$, and is solved for $s$ by a least squares approach represented by:

$$s = (\Delta \mathbf{x}^T \Delta \mathbf{x})^{-1} \Delta \mathbf{x}^T t \tag{7}$$

where superscript T indicates the transpose of a matrix (Del Pezzo and Giudicepietro, 2002).

Once $s_x$ and $s_y$ are solved for, they can be inverted to obtain the wave phase velocity components, $c_x$ and $c_y$, respectively. Additionally, the modeled delay times $t_m$ can be calculated by solving Eq. 6 for $t$. From $t_m$, we estimate the model error using root mean square error (RMSE) and normalized root mean square error (NRMSE):

$$RMSE = \sqrt{\frac{\sum_{i=1}^{N_s(N_s-1)/2}(t_{m,i} - t_i)^2}{N_s(N_s - 1)/2}} \tag{8}$$

$$NRMSE = \sqrt{\frac{\sum_{i=1}^{N_s(N_s-1)/2}(t_{m,i} - t_i)^2}{\sum_{i=1}^{N_s(N_s-1)/2}(t_i)^2}} \tag{9}$$

Events with sufficiently small RMSE and NRMSE in the modeled delay times can be considered "trackable" events in that there is higher confidence in the wave velocity estimates for those events. After processing multiple years of data from the Toronto and New York pressure sensor networks and analyzing the resulting RMSE and NRMSE distributions, it was found that a maximum RMSE threshold of 90 s and a maximum NRMSE threshold of 0.1 are reasonable to consider a wave event

trackable.

Additionally, we require that wave events be captured by at least 4 sensors to be considered trackable. While the slowness vector and corresponding error metrics can be calculated for events captured by only 3 sensors, the calculation is less constrained because there are only 3 delay times $\delta t_{opt}$ in the calculation (compared to 6 delay times for events captured by 4 sensors, 10 delay times for events captured by 5 sensors, etc.). The result is that events captured by only 3 sensors can have small RMSE and NRMSE by chance much more easily than events captured by 4 or more sensors. For each robust and trackable wave event, the mean amplitude was calculated by averaging the difference between the maximum and minimum values in the extracted event trace for each sensor which captured the event. The center period for the event was calculated as the mean of the wave period corresponding to $b_{max}$ for each sensor which captured the event.

## 4 Pressure disturbance examples

It is especially useful to have examples of wave events where other observational sources constrain the wave phase speed and direction. We discuss the following examples in this section, some of which have corroborating information on the wave phase speed and/or direction. The Lamb waves caused by the Hunga Tonga-Hunga Ha'apai volcanic eruption in January 2022 represent a case where the origin of the waves is known and the phase speed is known as a function of air temperature (Amores et al., 2022). A gravity wave train which passed over Toronto on 25 February 2022 occurred coincident with a surface cyclone 100 km distant but in local conditions of sparse radar echo. A wave event on 4 February 2022 is a case associated with an outflow boundary clearly captured by Doppler radar data from the WSR-88D radar located in Upton, NY. In another example, waves coincided with a cold front which passed over Toronto on 15 November 2020. The cold front's associated narrow rain band and Doppler velocity wave (Miller et al., 2022) can be identified in WSR-88D radar data from Buffalo, NY. We also describe a wake low associated with a long-lived mesoscale convective system (MCS) which passed over Long Island on 14 September 2021. These events were each manually chosen after roughly 40 months of pressure data were processed. The gravity wave train on 25 February 2022 and cold front example on 15 November 2020 are not unusual; several other gravity wave trains and pressure jumps due to cold front passages were found. The other three example cases are atypical events, however. It is extremely rare to detect a pressure signal due to a volcanic eruption several thousand km away. While other outflow boundaries were found to produce detectable pressure waves, the case on 4 February 2022 was unusual in terms of the time of year when it occurred (most other pressure waves associated with outflow boundary passages occurred during the warm season) and the radar signature. The wake low detected on 14 September 2021 was the only wake low event detected by our pressure sensor network.

### 4.1 15-17 January 2022: Hunga Tonga-Hunga Ha'apai eruption and Lamb wave

On 14-15 January, 2022, large eruptions occurred at the Hunga Tonga volcano in the south Pacific Ocean which produced ash plumes reaching the mesosphere and a series of shock waves. A particularly violent, submarine eruption occurred at around 0400 UTC 15 January (Global Volcanism Program, 2022). Satellite data suggest that the ash plume associated with this eruption reached as high as 57 km a.s.l. (Carr et al., 2022; Proud et al., 2022) and contained roughly 400 million kg of sulfur dioxide.

Damaging tsunami waves due to the eruption were observed as far as Peru (Global Volcanism Program, 2022). Subsequent analyses of the atmospheric pressure waves from the eruption have classified the pressure wave observed far from the source eruption as a Lamb wave (e.g., Amores et al., 2022). This Lamb wave has been the subject of several studies and media reports since the time of the eruption (e.g., Amores et al., 2022; Adam, 2022; Burt, 2022; Bhatia and Fountain, 2022; Vergoz et al., 2022). The pressure signal associated with the Lamb wave was observed to circle the Earth several times with estimated phase speeds exceeding $100 \text{ m s}^{-1}$ (Adam, 2022; Burt, 2022; Vergoz et al., 2022).

To identify and characterize the pressure waves from this event, we combined the three regional sensor networks (Fig. 2) to effectively create an array of 18 sensors which were active at the time of the Lamb wave passages. Table 1 summarizes events identified during this period which meet the robust event criteria outlined in the methods (captured by at least 4 sensors, RMSE $\leq$ 90 s, and NRMSE $\leq$ 0.1), in addition to having a mean optimal cross-correlation between the extracted event traces exceeding 0.75. The initial outbound (traveling from Tonga to the antipode location in Algeria) waves manifested as 3 separate detected events between 1509 and 1714 UTC 15 January, each with high wave frequencies (i.e., low wave periods barely over 1 minute) and low amplitudes (up to 0.3 hPa). The earliest, and strongest, of these events had a phase speed of $326.6 \text{ m s}^{-1}$ and direction of $64.2°$ (i.e., to the east-northeast). The subsequent rebound (traveling from the antipode location to Tonga) waves were detected as 2 separate events between 0356 and 0729 UTC 16 January. The rebound waves had a much higher amplitude (roughly 2.4 hPa) and lower wave frequency (i.e., longer wave periods on the order of roughly 10 minutes) than the outbound waves. The earliest rebound wave event had a phase speed of $292.1 \text{ m s}^{-1}$ and direction of $261.9°$ (i.e., to the west-southwest). Our networks captured the initial outbound waves from Tonga to the antipode at Algeria [Outbound (i), (ii), and (iii) in Table 1] and the first rebound waves back from Algeria [Rebound (i) and (ii) in Table 1]. Subsequent reverberations of the Lamb waves were not trackable with our sensors and methods by the above criteria due to a combination of the low amplitudes, high frequency, and large phase speeds confounding the process of approximating delay times between sensors (an issue also described by Grivet-Talocia and Einaudi, 1998).

## 4.2 25 February 2022: Gravity wave train over Toronto

Four pressure sensors in Toronto were used to detect and track a series of pressure oscillations between 0816 and 1359 UTC on 25 February 2022, with a particularly large pressure peak near 1000 UTC (Fig. 9). The mean amplitude of the event across the 4 sensors was 2.1 hPa, and the wave train was estimated to propagate at $45.4 \text{ m s}^{-1}$ at $73°$ (i.e., to the east-northeast). The center wave period was 00:20:22. At this time a mature surface cyclone was located roughly 100 km to the south of Toronto. Linear bands of reflectivity were identified in WSR-88D radar data from Buffalo, NY, in the hours leading up to the detected pressure waves, but between 0900 UTC and 1200 UTC there was only sparse radar echo over the Toronto area. Between 1230 and 1430 UTC there was radar echo across the Toronto area, and a set of Doppler velocity waves was identified from the WSR-88D data using the methods in Miller et al. (2022). Those Doppler velocity waves appeared to propagate toward the northeast, in roughly the same direction as the detected pressure waves (Video Supplement Animation-Figure-S01).

The nearest available sounding during the duration of the wave event appeared to indicate adequate environmental conditions for gravity wave ducting (Lindzen and Tung, 1976; Koch and O'Handley, 1997). The sounding from Buffalo, NY, valid at 1200

UTC on 25 Feb 2022, shows a temperature inversion roughly 2 km deep (Fig. 10), which serves as the "ducting layer" directly above a shallow boundary layer. A moist neutral or conditionally unstable layer (indicated by near-zero or negative values of the vertical gradient in equivalent saturation potential temperature) was above this inversion extending to around 4500 m MSL and serves as the "trapping layer" in Fig. 10. The sharp change in stability between the inversion layer and conditionally unstable layer at roughly 2800 m MSL could serve as a reflector of gravity wave energy. The apparent presence of a gravity wave duct during the detected pressure wave event raises confidence that the pressure waves were gravity waves.

### 4.3 4 February 2022: Outflow pressure jump and subsequent oscillations over Long Island

Between 1730 and 1900 UTC on 4 February 2022, an event with amplitude of roughly 1.8 hPa was detected by five pressure sensors in the New York City metro area and Long Island. This event was a positive jump in pressure followed, to varying degrees in each sensor, by weak oscillations in the pressure trace (Fig. 11). Prior to the jump in pressure, there had been a decreasing trend in the pressure traces for several hours. At the same time, WSR-88D weather radar data from Upton, NY, showed widespread precipitation echo over Long Island. A wave feature was apparent in the Doppler radial velocity data, which could also be identified following the methods of Miller et al. (2022) (Fig. 12 and Video Supplement Animation-Figure-S02). This wave event had a phase speed of 21.1 m s$^{-1}$ and direction of 118.2° clockwise from north (i.e., southeastward). The values are consistent with the radar-detected Doppler velocity wave feature (Video Supplement Animation-Figure-S02).

Operational one-minute Automated Surface Observing System (ASOS) data (Fig. 13a) also recorded a jump in the surface pressure of nearly 2 hPa. Near the time of this jump, there was also a peak in the wind speed and gusts, along with a brief shift in the wind direction from north-northeasterly to north-northwesterly (Fig. 13b). These features, along with the modest decrease in the temperature (Fig. 13a), are consistent with a convective outflow boundary (i.e., gust front). A "fine line" can be seen in WSR-88D reflectivity data at roughly the same location as the wave, which further suggests that a convective outflow was responsible for the pressure rise (Fig. 12 and Video Supplement Animation-Figure-S02).

### 4.4 15 November 2020: Cold front passage over Toronto

A robust and trackable wave event was detected by five pressure sensors in Toronto coincident with a cold front passage at roughly 2000 UTC on 15 November 2020. The pressure steadily dropped in the hours leading up to the frontal passage before abruptly rising 1-2 hPa as the cold air mass arrived. The pressure then dropped roughly 1 hPa about 30 min later. Some sensors recorded oscillations in the pressure trace embedded within the gradual pressure rise in the proceeding hours (Fig. 14). One-minute ASOS data from Buffalo, NY, also captured the pressure jump at roughly the same time as the temperature and dew point drop indicating the cold front passage (Fig. 15).

The pressure wave event for the cold front passage had an estimated phase speed of 27.5 m s$^{-1}$ at 65° (i.e., to the east-northeast), a mean amplitude of 1.8 hPa, and a center wave period of 00:02:08. WSR-88D radar data from Buffalo, NY, show a narrow band of high reflectivity and a Doppler velocity wave (identified following the methods in Miller et al., 2022) associated with the cold front advancing over Toronto at roughly 2000 UTC at a speed and direction consistent with the pressure wave (Video Supplement Animation-Figure-S03).

## 4.5 14 September 2021: Wake low associated with a mesoscale convective system

Between 0300 and 0400 UTC on 14 September 2021, a pressure drop of roughly 5 hPa and subsequent recovery occurred at four of the pressure sensors in New York City metro area and Long Island network (Fig. 16). This was detected as a wave event with an estimated propagation speed of 20.6 $m\ s^{-1}$ and propagation direction of 67.5° (i.e., to the east-northeast). ASOS data from Islip, NY (KISP; Fig. 17), and other stations in the area (not shown) also recorded the pressure minimum. This wave event occurred near the time of a mesoscale convective system (MCS) passage over Long Island as indicated by reflectivity data from the WSR-88D radar in Upton, NY (Fig. 18 and Video Supplement Animation-Figure-S04). In addition to the precipitation echo associated with the MCS translating from northwest to southeast, there was also a stationary region of weak echo with low dual-polarization correlation coefficient (shown in greyscale following Tomkins et al., 2022) in the vicinity of the radar. The stationary weak echo was likely non-meteorological and due to either birds or insects. The precipitation echo appears to be entirely past KISP by 0342 UTC (Fig. 18c and Video Supplement Animation-Figure-S04), which is roughly the same time as the minimum pressure at KISP (Fig. 17a).

This pressure minimum appears to be consistent with a *wake low*, associated with subsidence heating in the rear inflow jet (Markowski and Richardson, 2010; Johnson and Hamilton, 1988). The subsidence heating does not necessarily lead to warming at the surface, which was not observed in the ASOS data (Fig. 17a), but decreased air density aloft due to warming will still lead to a surface pressure decrease. Markowski and Richardson (2010) also note that a property of wake lows associated with a translating squall line is that the center of convergence due to the wake low does not perfectly align with the center of the wake low. Rather, the convergence center slightly lags behind the wake low center. In the 14 September 2021 example, the ASOS time series data have a wind speed minimum co-occurring with a shift in the wind direction from near 100° (east-southeasterly) to near 280° (west-northwesterly) which can be interpreted as the convergence maximum (Fig. 17b). This convergence maximum occurs slightly after the pressure minimum associated with the wake low (Fig. 17a), consistent with the Markowski and Richardson (2010) description.

## 5 Discussion and Summary

In this study, a wavelet-based method was used to identify wave events in time series pressure data from networks of high precision sensors (0.8 Pa noise floor) recording the pressure every second. In addition to identifying wave events in each sensor individually, the delay times in wave passage among sensors in a given network were used to determine the direction of wave propagation and phase velocity. The methods shown are intended mainly for post-processing of pressure data for research applications, and not for real time, operational use. A benefit to this method is that it can be fully automated to detect wave events across many months of data, and we have made the processing code openly available (Allen and Miller, 2023).

Overall, the method was most successful at tracking pressure wave events with relatively large amplitudes (on the order of 0.3 hPa or more) and longer periods (i.e., lower frequencies; on the order of 5 minutes or more). Low-amplitude, high-frequency waves likely propagated across the sensor networks many times, but these waves were difficult to reconstruct, due to their wavelet signal being weaker, and to track, due to aliasing of the waveform conflating the time lag estimates between

sensors. We use a rather strict criteria for detecting a wave event in a given sensor; the peak wavelet power must exceed 10 times the mean value for a given wave period (i.e., K = 10; eq. 4). Grivet-Talocia and Einaudi (1998) also used wavelet analysis to detect pressure waves with a scale dependent threshold; their K value was only 2. Therefore, many wave events detected using our criteria will be relatively high amplitude (on the order of 1 hPa or more), including most examples shown in Sect. 4. Depending on the desired application, this threshold and the other thresholds used in the wave detection can be adjusted.

Environmental factors can influence whether or not a given gravity wave is detected by our surface pressure sensors. Gravity waves aloft will not always produce a detectable pressure signal at the surface, for example if the planetary boundary layer is neutral or unstable (e.g., Kjelaas et al., 1974). Another possible limitation is that in their current network deployments the pressure sensors are too far apart to track highly localized disturbances, particularly for the New York City/Long Island sensor network. Our method may not always properly track waves which are modified by local conditions (which may alter their amplitude, frequency, and/or phase velocity) as they propagate across the sensor network. Future work will examine data from networks of pressure sensors a few km to a few m apart and the degree to which signals associated with waves in shallow marine clouds are detectable with these sensors.

Deployment of networks of low-cost, high-precision sensors opens myriad opportunities for monitoring the direction and speed of gravity waves that have not been previously available with conventional pressure sensors on operational weather stations due to their longer measurement interval and larger station spacing. A forthcoming publication will describe a 3+ year climatology of wave events detected by the pressure sensors deployed in New York and Toronto and address hypotheses regarding the relationship between gravity waves and local enhancements in snowfall rate within winter storms (i.e., snow bands). There are observational case studies demonstrating this connection (e.g., Bosart et al., 1998; Gaffin et al., 2003), but a multi-year data set with continuously-monitoring pressure sensors in context of radar data will enable a more comprehensive examination of the co-occurrence, or lack thereof, of gravity waves with snow bands across many winter storms.

*Code and data availability.* Data: The pressure time series data used throughout this publication can be found at https://doi.org/10.5281/zenodo.8136536 (Miller and Allen, 2023). The NWS NEXRAD Level-II data used in Figs. 12 and 18 can be accessed from the National Centers for Environmental Information (NCEI) at https://www.ncei.noaa.gov/products/radar/next-generation-weather-radar (NOAA National Weather Service Radar Operations Center, 1991). The NWS ASOS surface station data used to create Figs. 13 and 17 can be accessed from NCEI at https://www.ncei.noaa.gov/products/land-based-station/automated-surface-weather-observing-systems (NOAA National Centers for Environmental Information, 2021a). The radiosonde data used to create Fig. 10 can be accessed from NCEI at https://www.ncei.noaa.gov/products/weather-balloon/integrated-global-radiosonde-archive (NOAA National Centers for Environmental Information, 2021b).

Code: The code used for processing the pressure time series data can be found at https://doi.org/10.5281/zenodo.10150876 (Allen and Miller, 2023).

*Video supplement.* List of animations with captions and filenames

All animations can be viewed at: https://doi.org/10.5446/s_1476. Individual animations can be viewed by following the DOI URL.

Animation-Figure-S01: Animated maps of (a) reflectivity and (b) Doppler velocity wave detection for NWS WSR-88D radar data from Buffalo, NY, at 0.5° tilt, from 0706 UTC to 1457 UTC on 25 Feb 2022. In (a), reflectivity values are shown in greyscale when there is likely enhancement due to melting (Tomkins et al., 2022). Filled blue circles indicate locations of
pressure sensors which captured the wave event described in Sect. 4.2, and unfilled blue circles indicate locations of pressure sensors which did not capture the wave event. Title: 25 Feb 2022 KBUF Reflectivity and Doppler Velocity Waves. DOI: https://doi.org/10.5446/62539

Animation-Figure-S02: Animated maps of (a) reflectivity and (b) Doppler velocity wave detection for NWS WSR-88D radar data from Upton, NY, at 0.5° tilt, from 1541 UTC to 2129 UTC on 4 Feb 2022. In (a), reflectivity values are shown
in greyscale when there is likely enhancement due to melting (Tomkins et al., 2022). Filled blue circles indicate locations of pressure sensors which captured the wave event described in Sect. 4.3, and unfilled blue circles indicate locations of pressure sensors which did not capture the wave event. Goes with Fig. 12. Title: 04 Feb 2022 KOKX Reflectivity and Doppler Velocity Waves. DOI: https://doi.org/10.5446/62540

Animation-Figure-S03: Animated maps of (a) reflectivity and (b) Doppler velocity wave detection for NWS WSR-88D
radar data from Buffalo, NY, at 0.5° tilt, from 1805 UTC to 2324 UTC on 15 Nov 2020. In (a), reflectivity values are shown in greyscale when there is likely enhancement due to melting (Tomkins et al., 2022). Filled blue circles indicate locations of pressure sensors which captured the wave event described in Sect. 4.4, and unfilled blue circles indicate locations of pressure sensors which did not capture the wave event. Title: 15 Nov 2020 KBUF Reflectivity and Doppler Velocity Waves. DOI: https://doi.org/10.5446/62541

Animation-Figure-S04: Animated maps of (a) reflectivity and (b) Doppler velocity wave detection for NWS WSR-88D radar data from Upton, NY, at 0.5° tilt, from 0003 UTC to 0727 UTC on 14 Sep 2021. In (a), reflectivity values are shown in greyscale when there is likely enhancement due to melting (Tomkins et al., 2022). Filled blue circles indicate locations of pressure sensors which captured the wave event described in Sect. 4.5, and unfilled blue circles indicate locations of pressure sensors which did not capture the wave event. Goes with Fig. 18. Title: 14 Sep 2021 KOKX Reflectivity and Doppler Velocity
Waves. DOI: https://doi.org/10.5446/62542

*Author contributions.* LRA, SEY, and MAM conceptualized the project and designed the methodology. MAM designed and built the pressure sensors and managed the pressure sensor networks. LRA and MAM wrote the software. LRA and LMT created the visualizations with input from SEY. LRA prepared the manuscript, SEY edited the manuscript, and MAM and LMT contributed to the final stages of reviewing and editing.

*Competing interests.* The contact author has declared that none of the authors have any competing interests.

*Acknowledgements.* The authors express their sincere appreciation to the pressure sensor hosts in Toronto, New York City metro area and Long Island, and Raleigh, including colleagues from Environment Canada, Stony Brook University, Columbia University, and friends, including Jase Bernhardt, Drew Claybrook, Brian Colle, Daniel Horn, Daniel Michelson, Robert Pincus, Spencer Rhodes, Adam Sobel, David Stark, and Jeff Waldstreicher, who graciously agreed to plug in pressure sensors to their home internet. Figure 10 was made using code developed by Kevin Burris. The development and refinement of the methodology benefited from discussions with DelWayne Bohnenstiehl, Brian Colle, Brian Mapes, Matthew Parker, and Minghua Zhang.

This work was supported by the National Science Foundation (AGS-1905736), the National Aeronautics and Space Administration (80NSSC19K0354), the Office of Naval Research (N000142112116), and the Center for Geospatial Analytics at North Carolina State University.

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

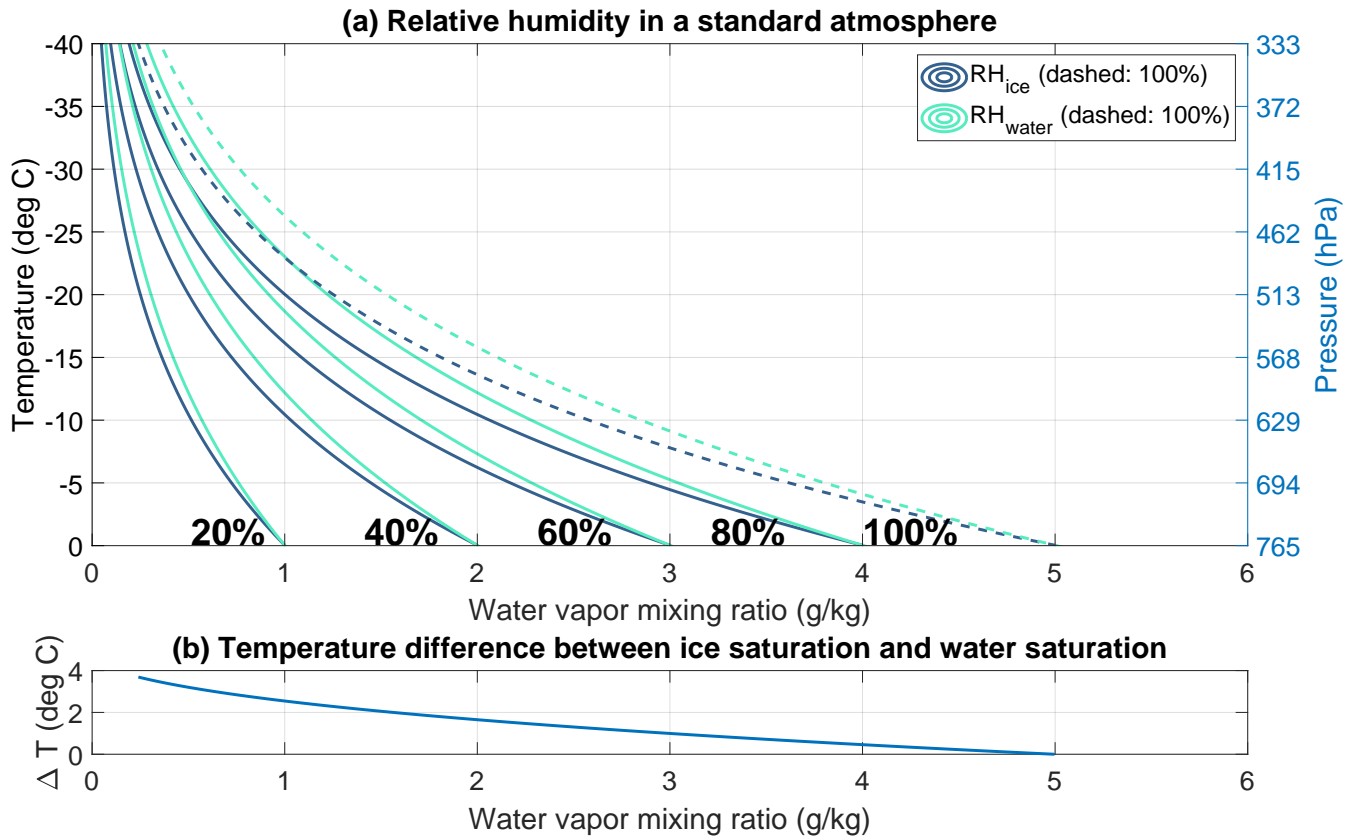

**Figure 1.** (a) Contours of relative humidity with respect to ice ($RH_{ice}$) and with respect to liquid water ($RH_{water}$) as a function of temperature and water vapor mixing ratio, assuming a standard atmosphere temperature-pressure relationship (corresponding pressure values on right axis). RH values are contoured at 20% intervals, with the 100% contour dashed. (b) The temperature difference between the 100% $RH_{ice}$ and 100% $RH_{water}$ contours for each water vapor mixing ratio.

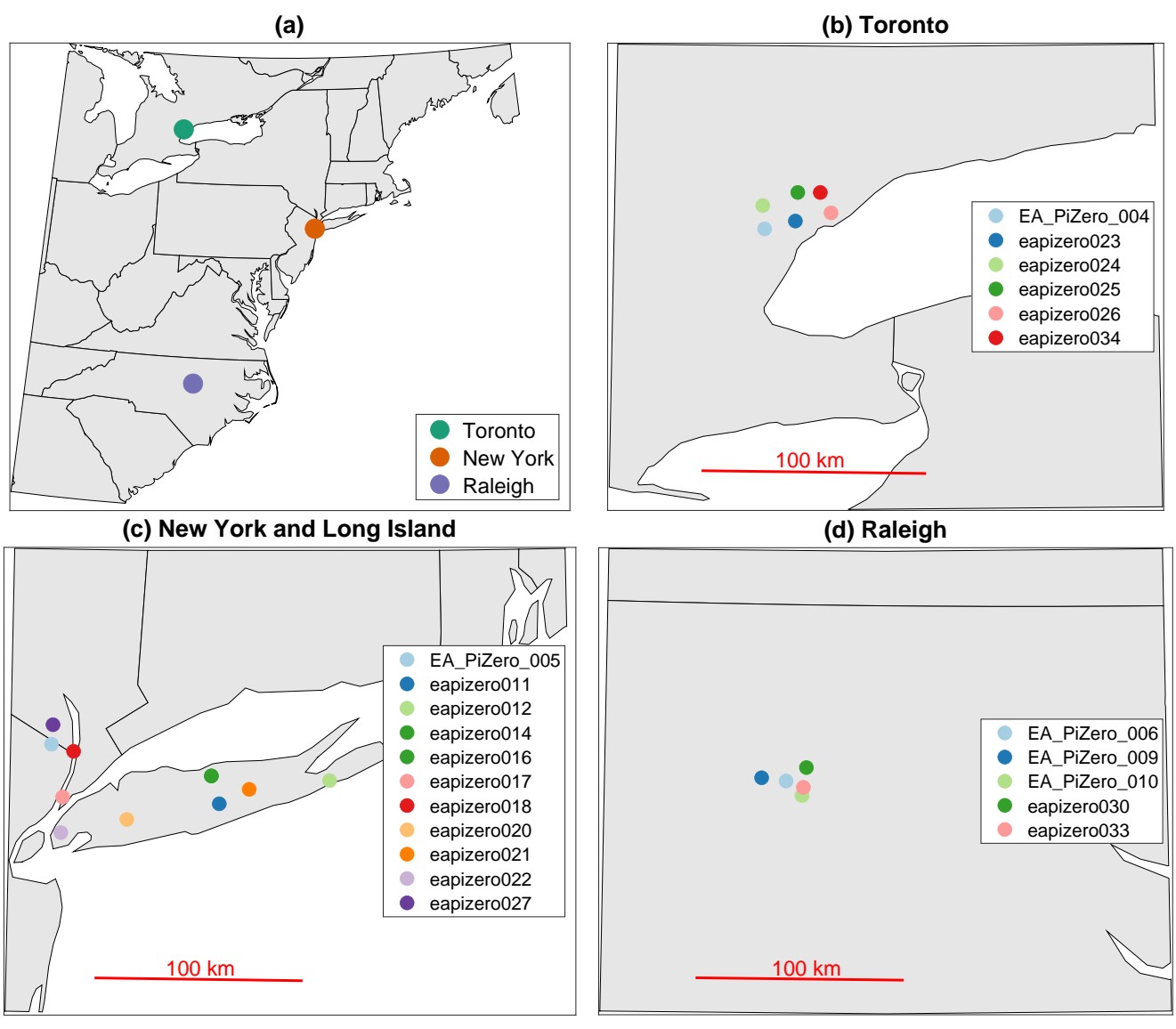

**Figure 2.** Locations of pressure sensor networks. (a) US northeast regional map with the locations of the Toronto, New York City and Long Island, and Raleigh networks indicated. Detailed maps of sensor locations in (b) Toronto, (c) New York and Long Island, and (d) Raleigh.

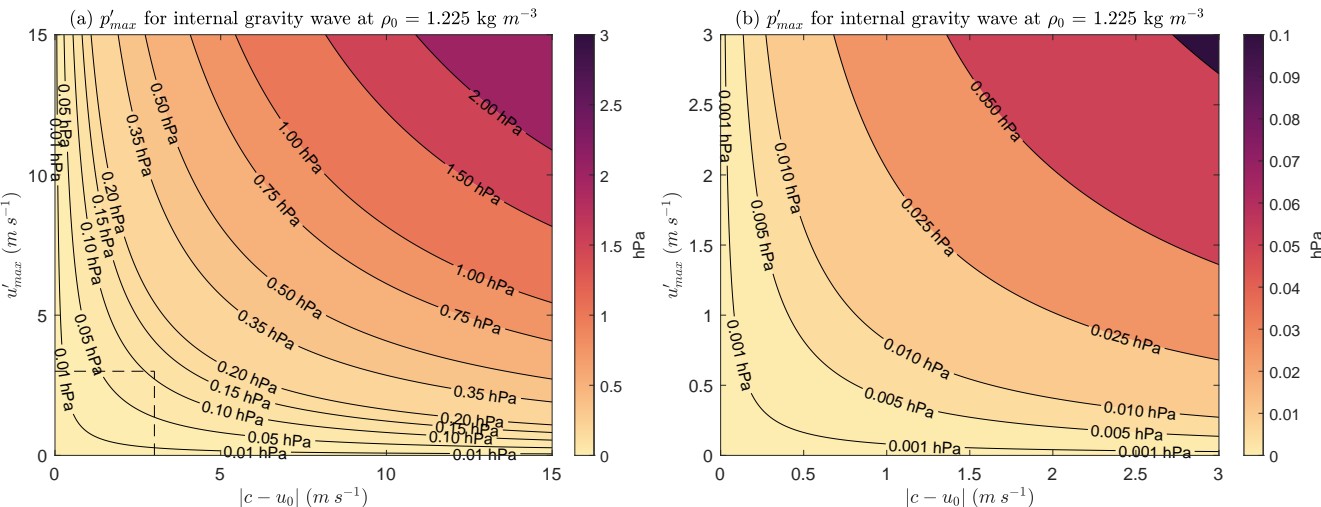

**Figure 3.** The maximum pressure perturbation $p'_{max}$ (hPa) contoured and colored as a function of the maximum velocity perturbation $u'_{max}$ (m s$^{-1}$) and absolute value of the difference between the wave phase speed and background wind speed $|c - u_0|$ (m s$^{-1}$) at a density $\rho_0$ of 1.225 $kg\ m^{-3}$, according to Eq. 2. In (a), $u'_{max}$ and $|c - u_0|$ up to 15 m s$^{-1}$ are shown. In (b), $u'_{max}$ and $|c - u_0|$ up to 3 m s$^{-1}$ are shown. The color scales differ in (a) and (b).

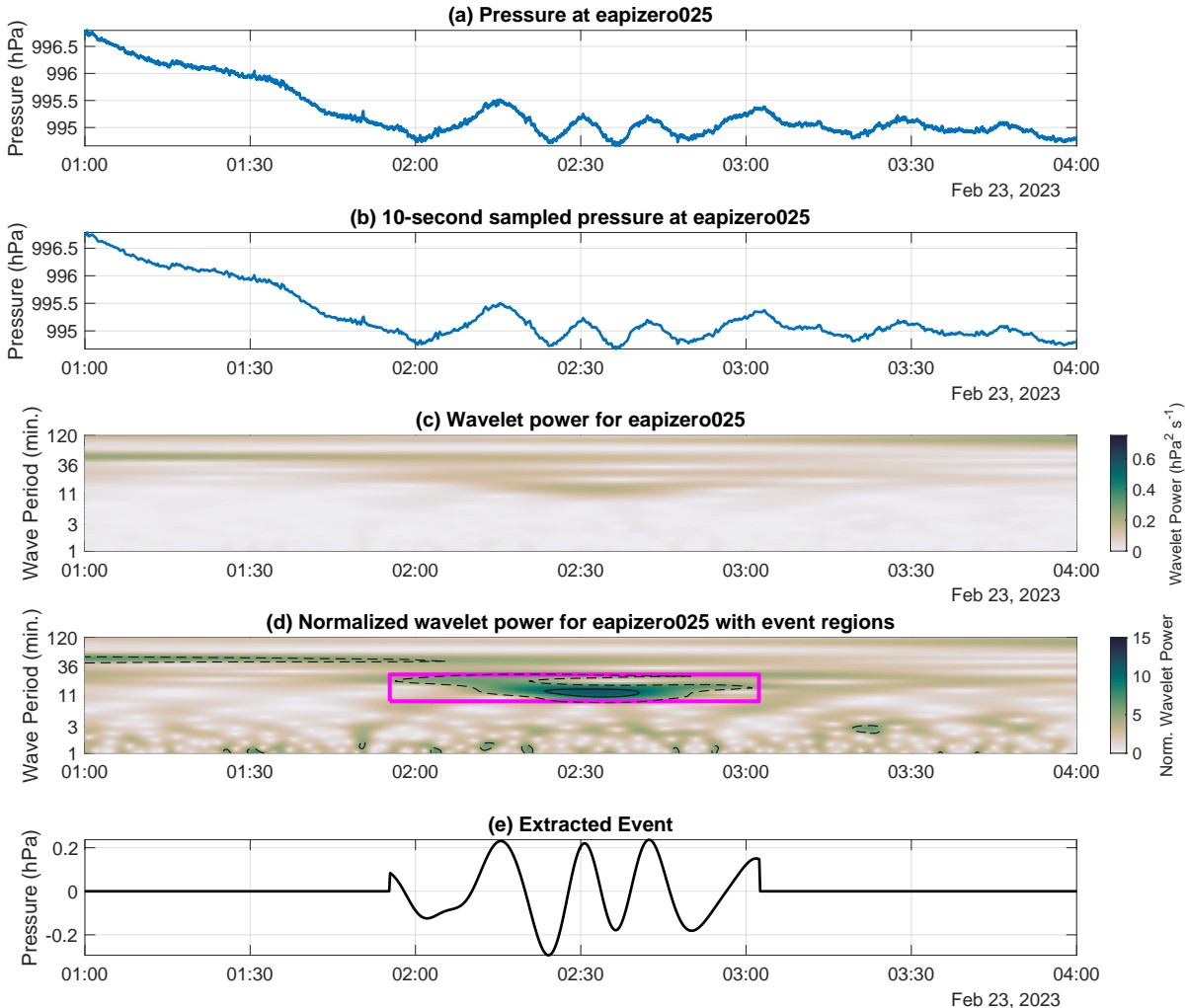

**Figure 4.** Steps in process of identifying an event corresponding to a gravity wave train passage on 23 Feb 2023 in sensor 25. (a) Original 1-second pressure time series. (b) 10-second moving average of the pressure time series, with every 10th point kept. (c) Wavelet power corresponding to the pressure time series in (b). (d) Wavelet power normalized by the mean for each wave period corresponding to the pressure trace, with contours for values of 5 (dashed) and 10 (solid). The event region corresponding to the wave described in the text is outlined in magenta in (d). (e) Time series of the extracted wave event. Wave periods in (c) and (d) are shown on a logarithmic scale. All times are UTC.

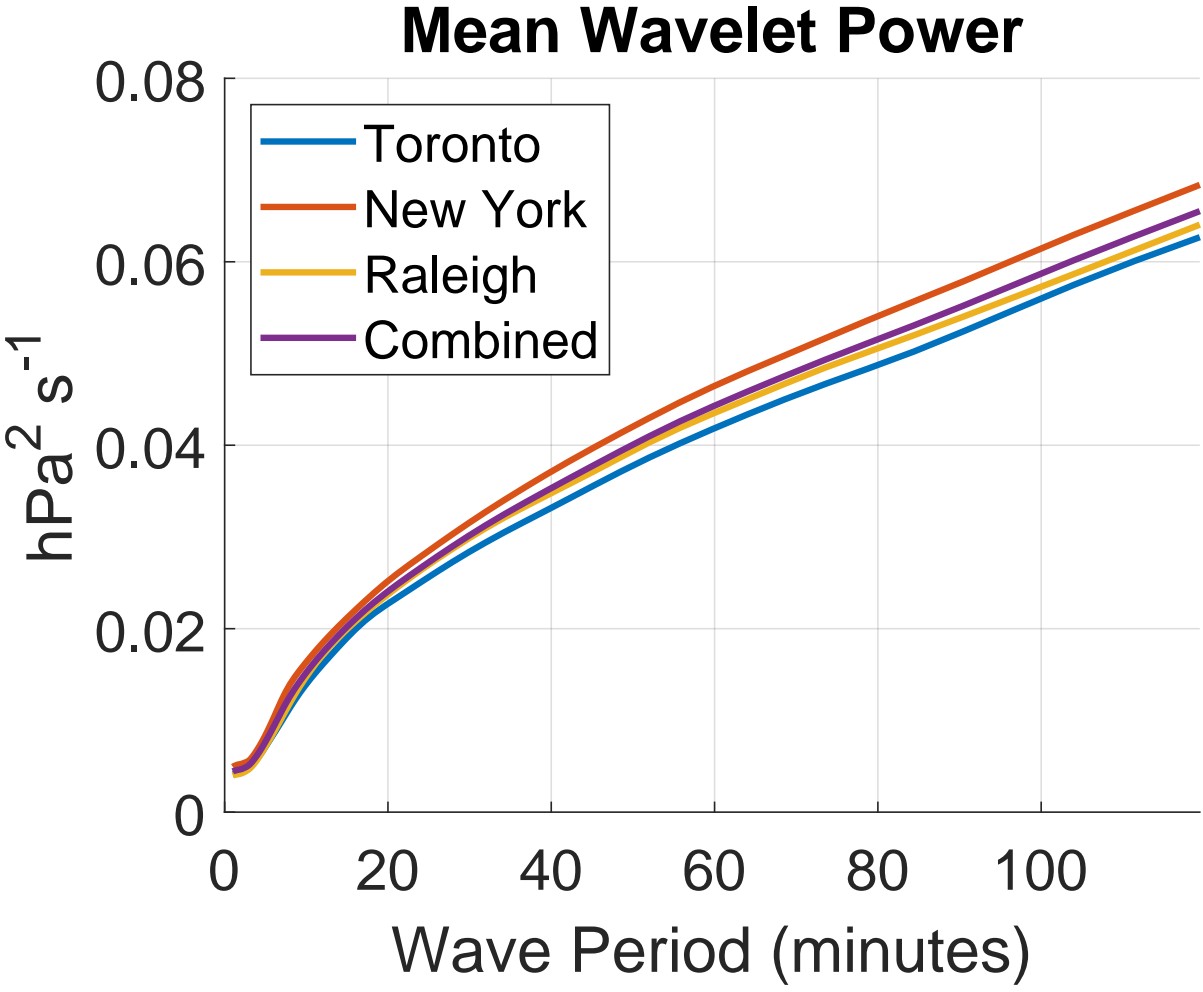

**Figure 5.** Mean wavelet power as a function of wave period across the entire data set (purple curve), and for the individual sensor networks around Toronto, ON, Canada (blue), New York City and Long Island, NY (red), and Raleigh, NC (yellow).

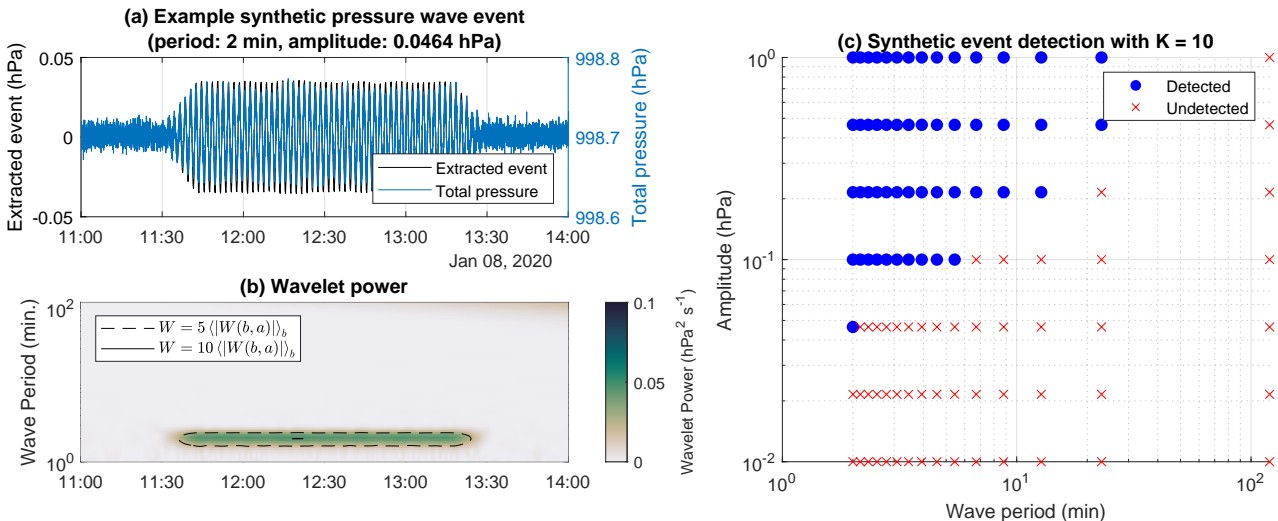

**Figure 6.** (a) Part of the synthetic time series of pressure data (blue) with an extracted wave event (black). This synthetic wave event had a wave period of 2 min and amplitude of $\pm 0.0464$ hPa. (b) The wavelet power corresponding to the time series shown in (a). The dashed and solid contours indicate where the wavelet power is 5 times and 10 times the mean wavelet power for a given wave period, respectively. (c) The synthetic wave events which were detected (blue filled circles) and undetected (red X symbols) using our methodology with a $K$ value of 10, as a function of their wave period and amplitude.

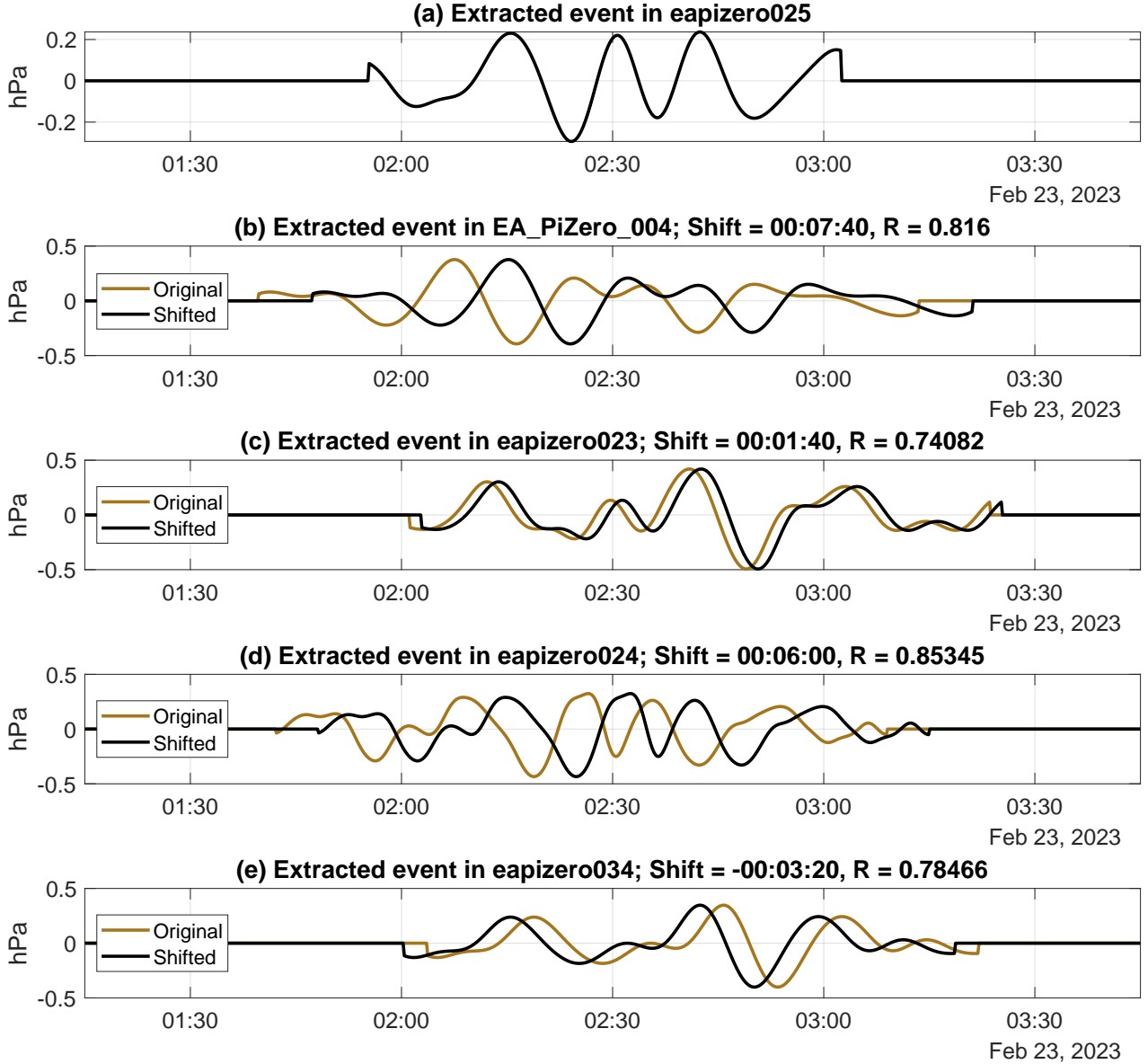

**Figure 7.** Extracted waveforms corresponding to the gravity wave train on 23 Feb, 2023 for sensors 25 (a), 04 (b), 23 (c), 24 (d), and 34 (e). In (b), (c), (d), and (e), brown lines show the extracted wave event with no time shift, and black lines show the extracted wave event shifted in time according to the peak cross-correlation with the extracted wave event time series for the event in sensor 25 (time shift and correlation coefficients are shown in subplot titles). All times are UTC.

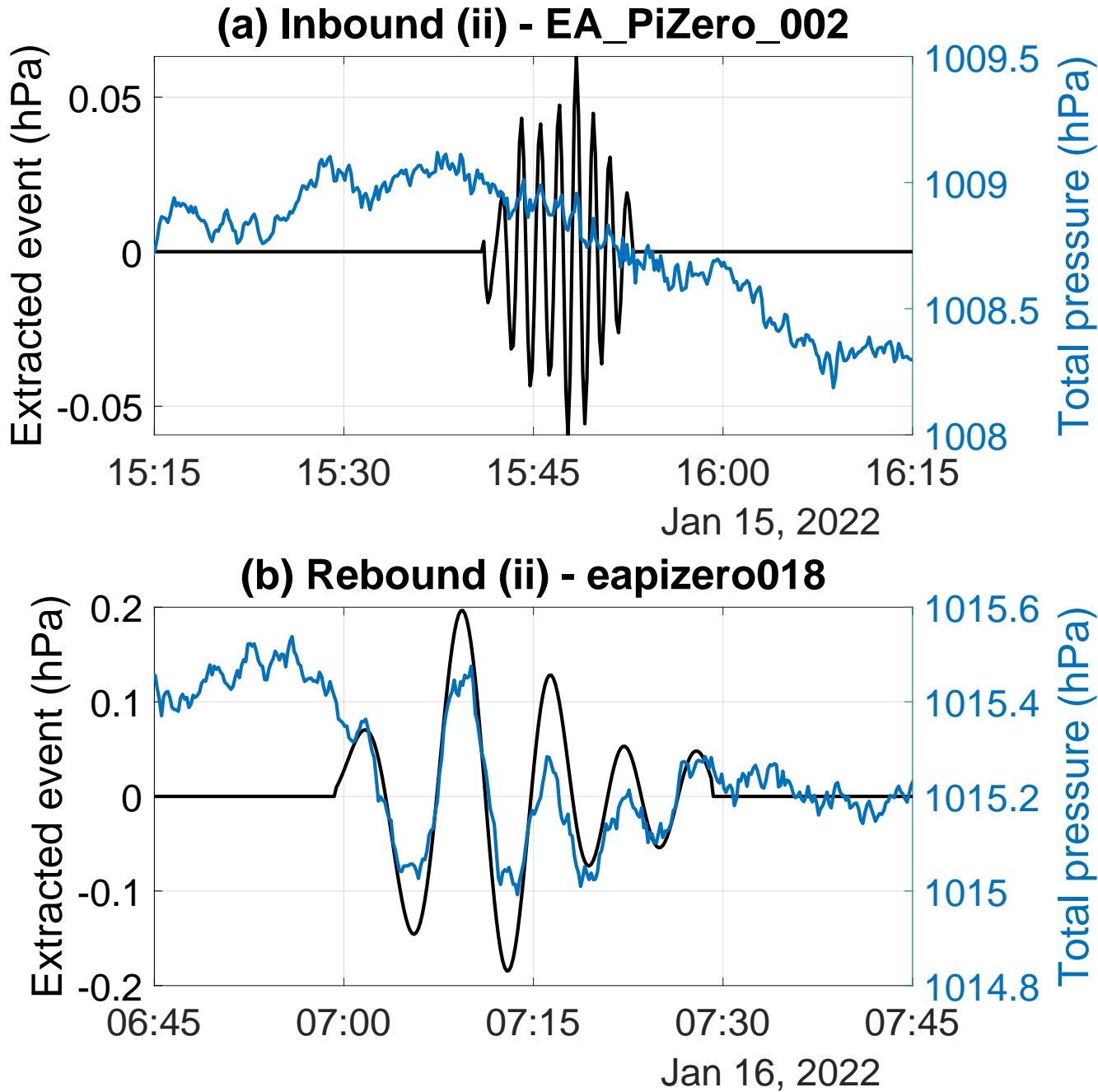

**Figure 8.** Pressure traces and extracted waveforms for two events associated with the Lamb waves caused by the Hunga Tonga-Hunga Ta'apai eruption on 15 Jan 2022. Extracted waveforms (black lines) are overlaid on the total pressure time series (blue lines). (a) the Outbound (ii) wave event propagating from Tonga toward the antipode location in Algeria in Sensor 02 (Table 1, column 3). (b) the Rebound (ii) wave event propagating from the antipode location to Tonga in Sensor 18 (Table 1, column 6). All times are UTC.

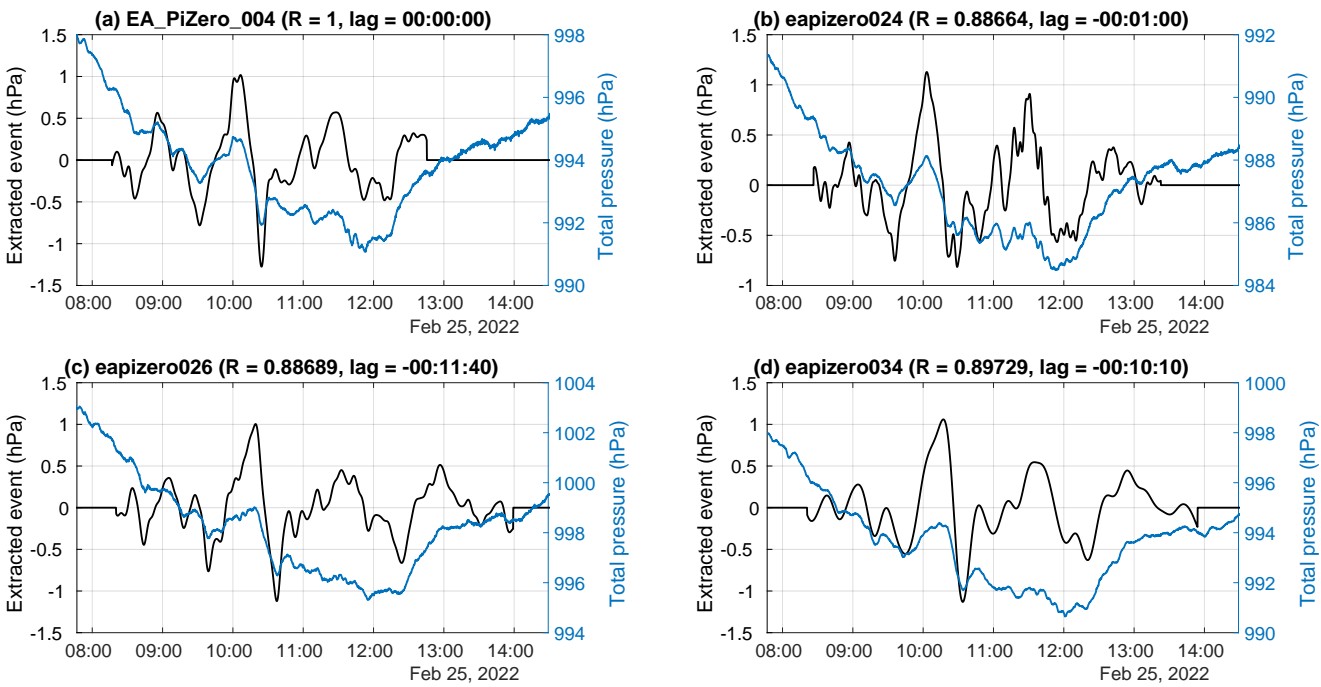

**Figure 9.** Pressure traces and extracted waveforms for the 25 Feb 2022 wave event in sensors (a) 04, (b) 24, (c) 26, and (d) 34. Extracted waveforms (black lines) are overlaid on the total pressure time series (blue lines). All times are UTC. Cross-correlations and lag times are indicated relative to sensor 04. Cross-correlations are computed for each variation of pairs of sensors (not shown).

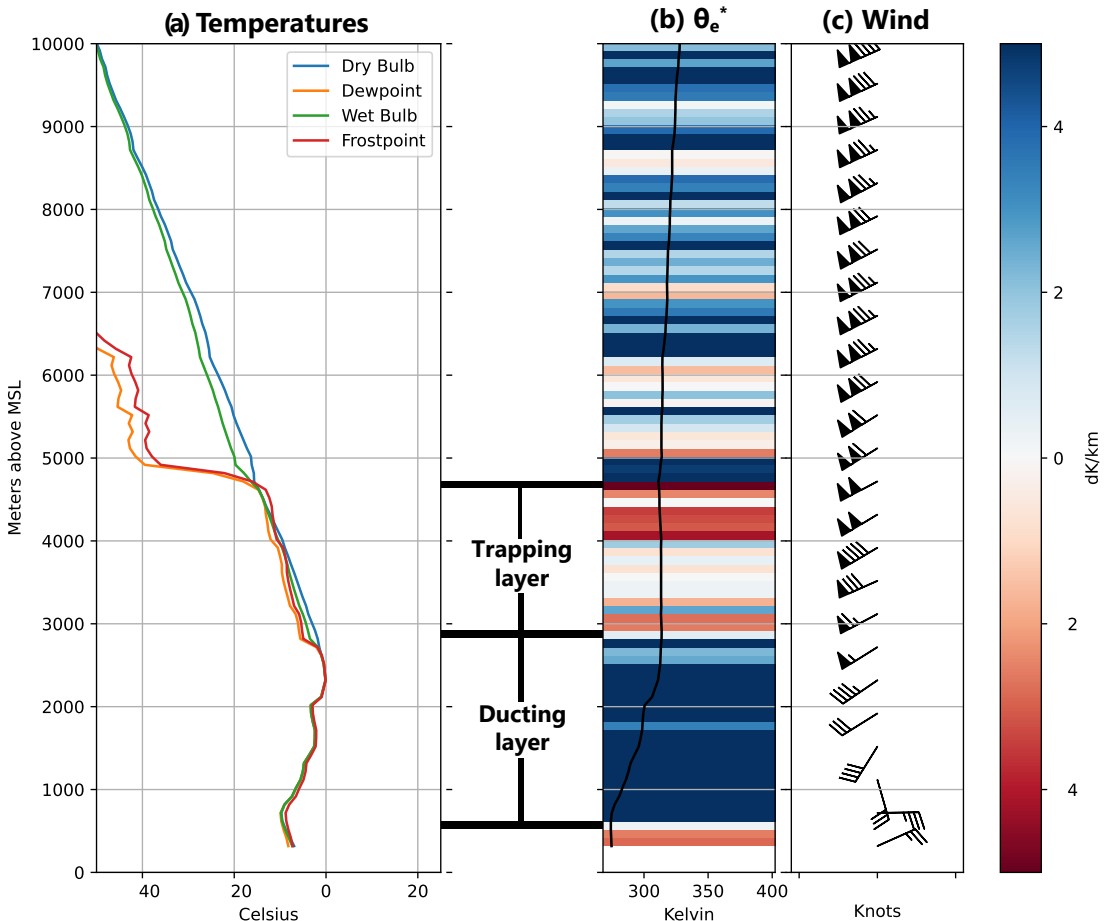

**Figure 10.** Upper air sounding data from Buffalo, NY, valid for 1200 UTC 25 Feb 2022. (a) Dry bulb temperature (blue), dew point (orange), wet bulb temperature (green), and frost point (red) profiles (all in °C). (b) Equivalent saturation potential temperature ($\theta_e^*$) profile (black line, K) overlaid on the vertical gradient in $\theta_e^*$ (K km$^{-1}$). Positive values (blue) of the vertical gradient in $\theta_e^*$ indicate absolute stability, while negative values (red) indicate conditional or absolute instability. (c) Horizontal wind profile (barbs, kts; colored according to wind speed). Annotation indicates the vertical extents of a ducting layer and a trapping layer according to the gravity wave duct criteria described by Lindzen and Tung (1976) and Koch and O'Handley (1997).

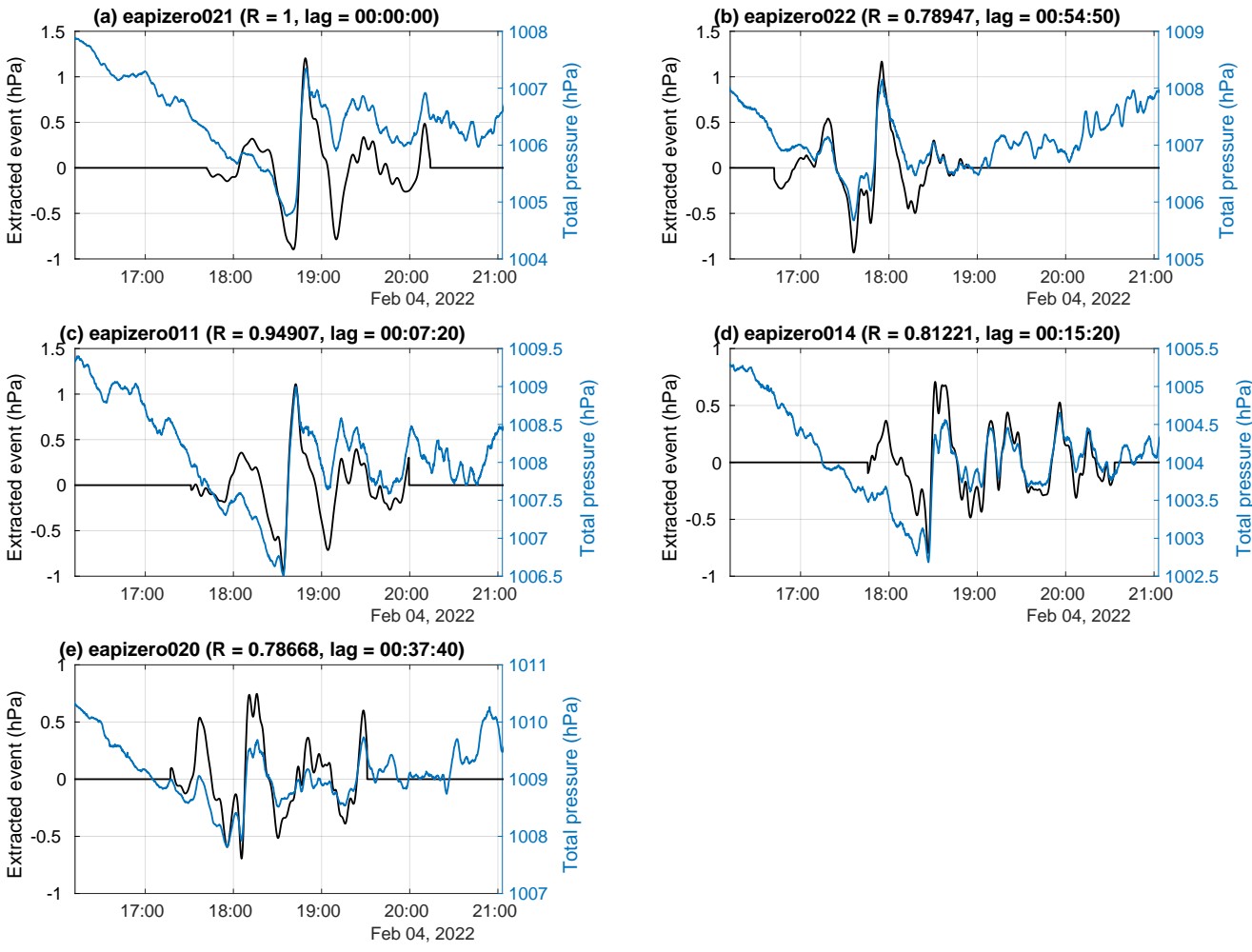

**Figure 11.** As in Fig. 9, but for the 04 Feb 2022 wave event in sensors (a) 21, (b) 22, (c) 11, (d) 14, and (e) 20, with cross-correlations and lag times indicated relative to sensors 21. Extracted waveforms (black lines) are overlaid on the total pressure time series (blue lines).

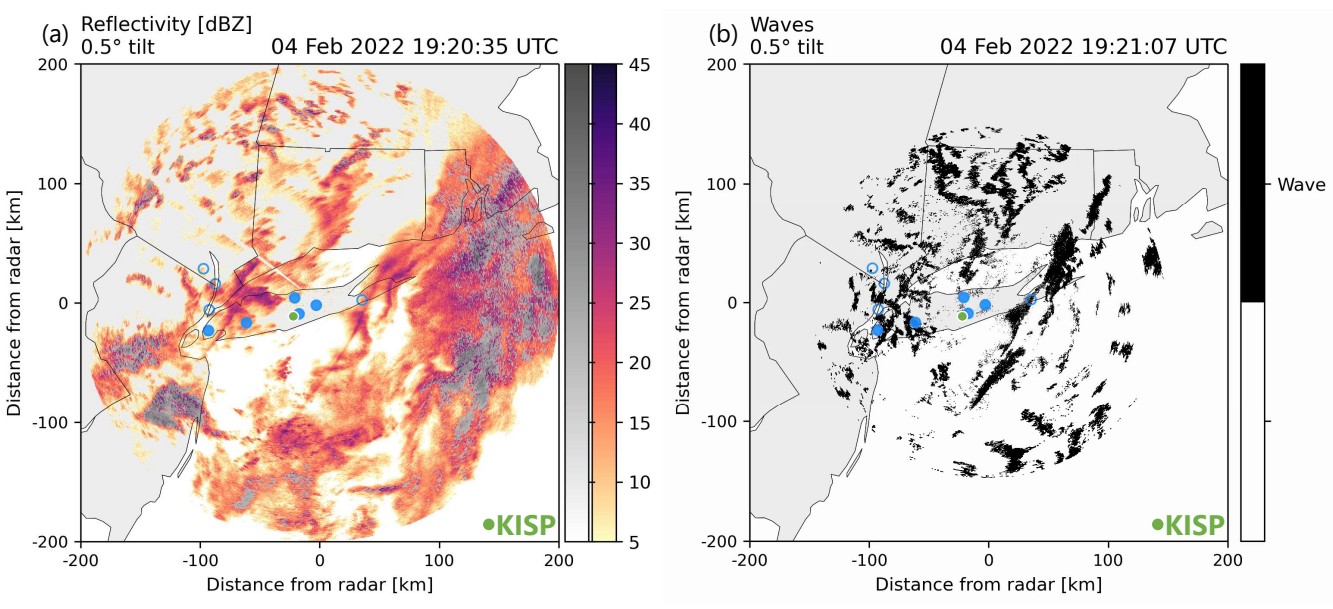

**Figure 12.** (a) Reflectivity and (b) Doppler velocity wave detection for NWS WSR-88D radar data from Upton, NY, at 0.5° tilt, at 1920 UTC on 4 Feb 2022. In (a), reflectivity values are shown in greyscale where there is likely enhancement due to melting (Tomkins et al., 2022). Filled blue circles indicate locations of pressure sensors which captured the wave event described in Sect. 4.3, and unfilled blue circles indicate locations of pressure sensors which did not capture the wave event. Filled green circle indicates location of Islip, NY, ASOS station (KISP). An animation of this figure showing the time sequence from 1541 to 2130 UTC is in Video Supplement Animation-Figure-S02.

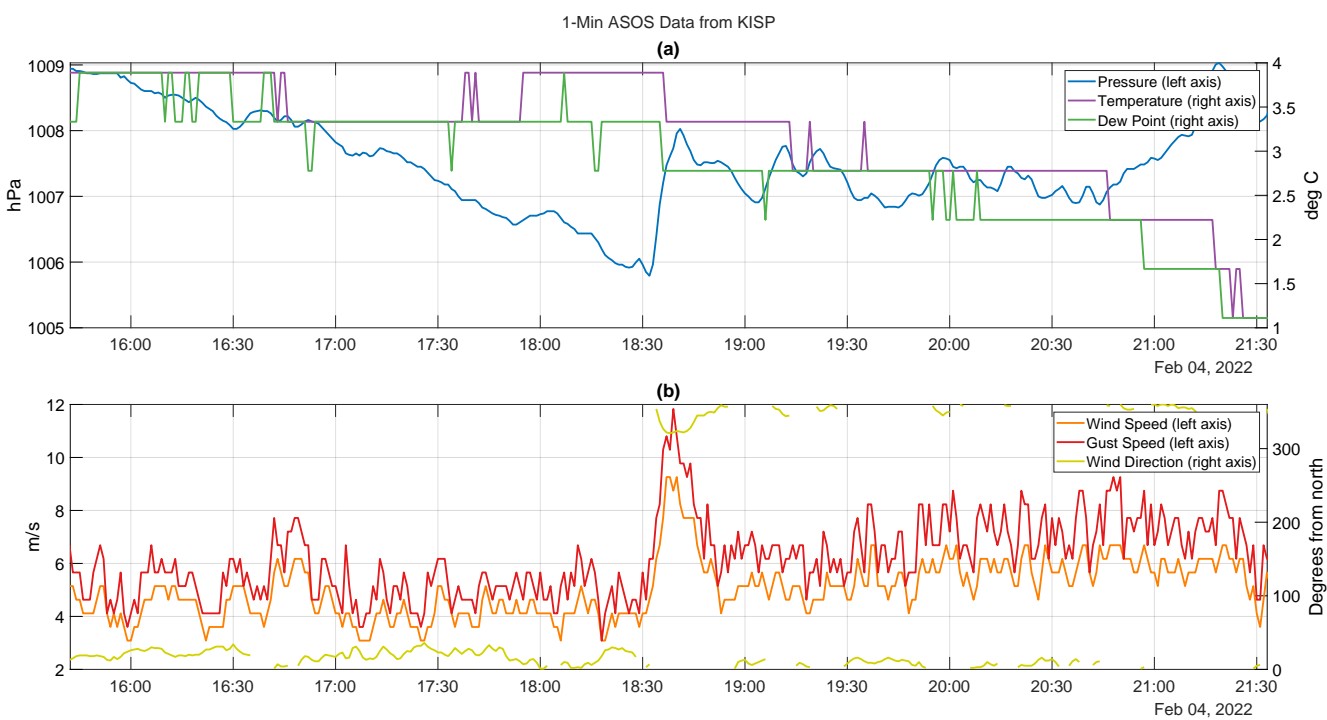

**Figure 13.** Time series of one-minute ASOS data from Islip, NY (KISP), on 4 February 2022. (a) Temperature (purple) dew point (red), and pressure (blue). (b) Wind speed (orange), wind gust speed (red), and wind direction in degrees clockwise from northerly (yellow). Wind direction is not plotted when it changes by more than $180°$ in consecutive observations (e.g., when crossing $0°$ or $360°$) or when the wind speed is below $1.5 \text{ m s}^{-1}$. All times are UTC.

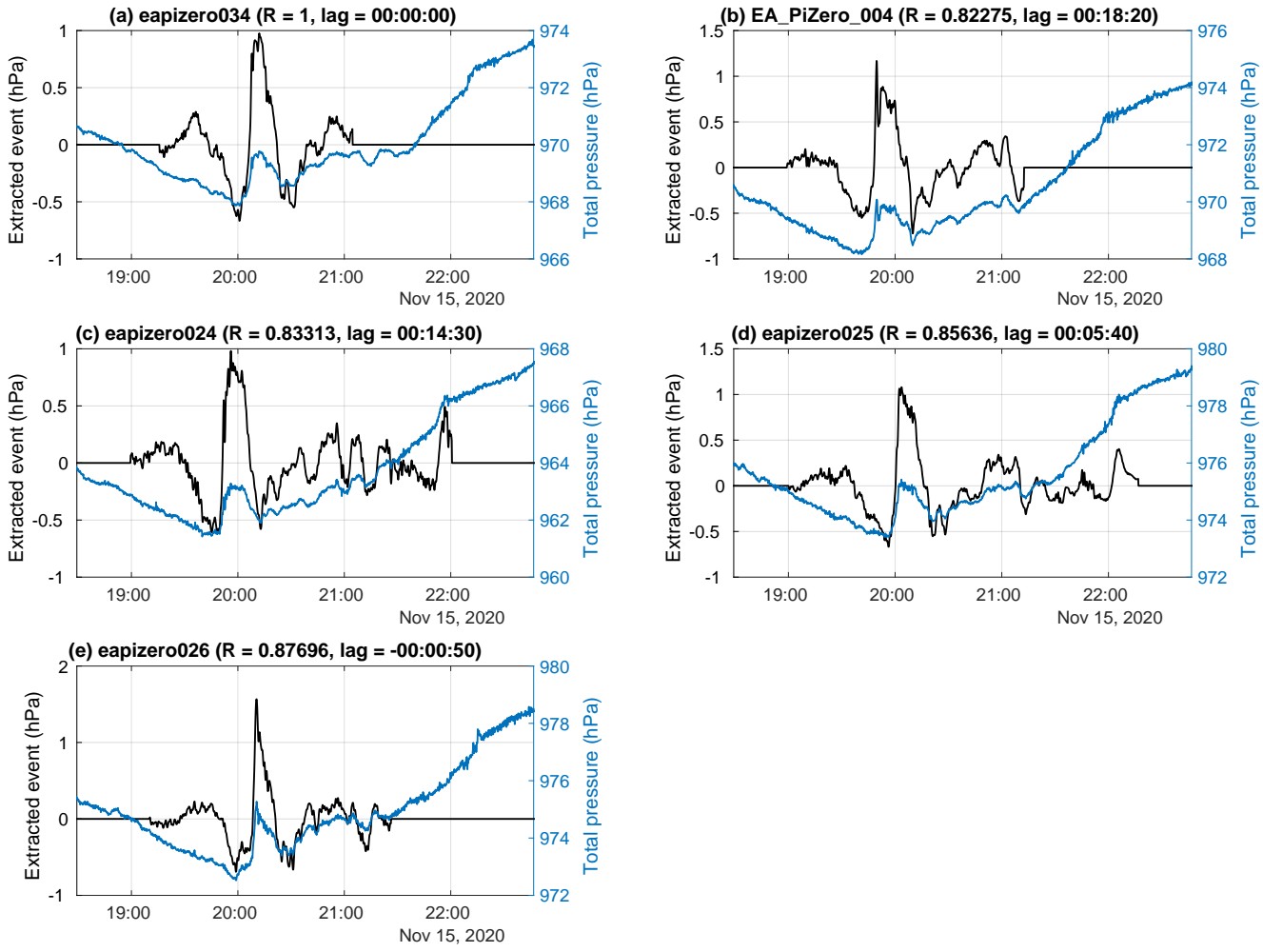

**Figure 14.** As in Fig. 9, but for the 15 Nov 2020 wave event in sensors (a) 34, (b) 04, (c) 24, (d) 25, and (e) 25, with cross-correlations and lag times indicated relative to sensor 34. Extracted waveforms (black lines) are overlaid on the total pressure time series (blue lines).

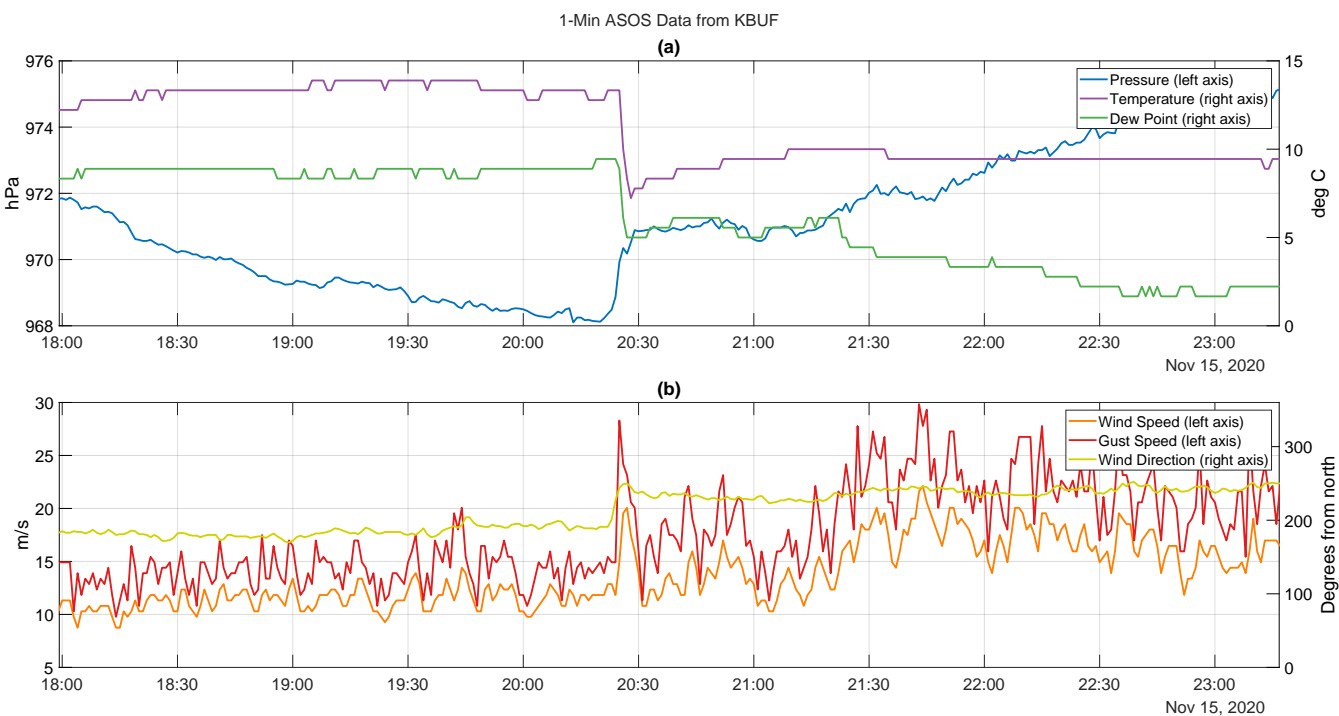

**Figure 15.** Time series of one-minute ASOS data from Buffalo, NY (KBUF), on 15 November 2020. (a) Temperature (purple) dew point (red), and pressure (blue). (b) Wind speed (orange), wind gust speed (red), and wind direction in degrees clockwise from northerly (yellow). Wind direction is not plotted when it changes by more than 180° in consecutive observations (e.g., when crossing 0° or 360°) or when the wind speed is below 1.5 m s$^{-1}$. All times are UTC.

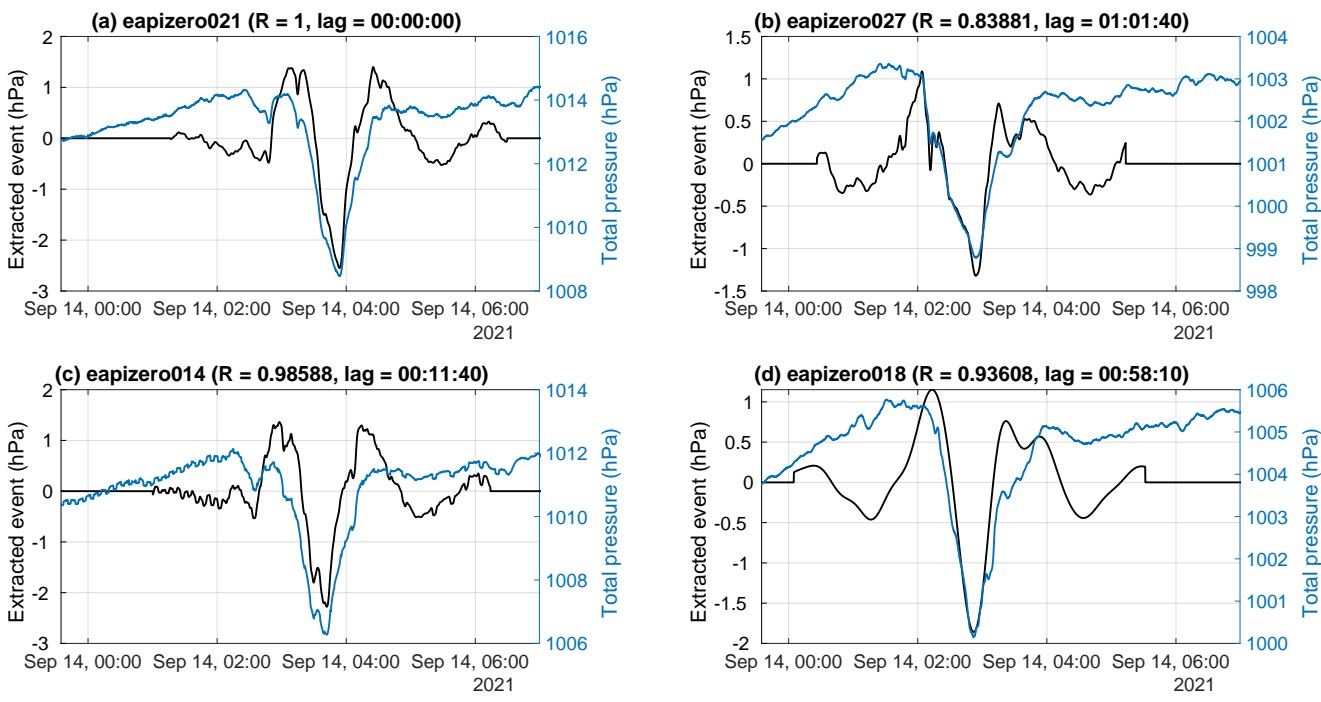

**Figure 16.** As in Fig. 9, but for the 14 Sep 2021 wave event in sensors (a) 21, (b) 27, (c) 14, and (d) 18, with cross-correlations and lag times indicated relative to sensor 21. Extracted waveforms (black lines) are overlaid on the total pressure time series (blue lines).

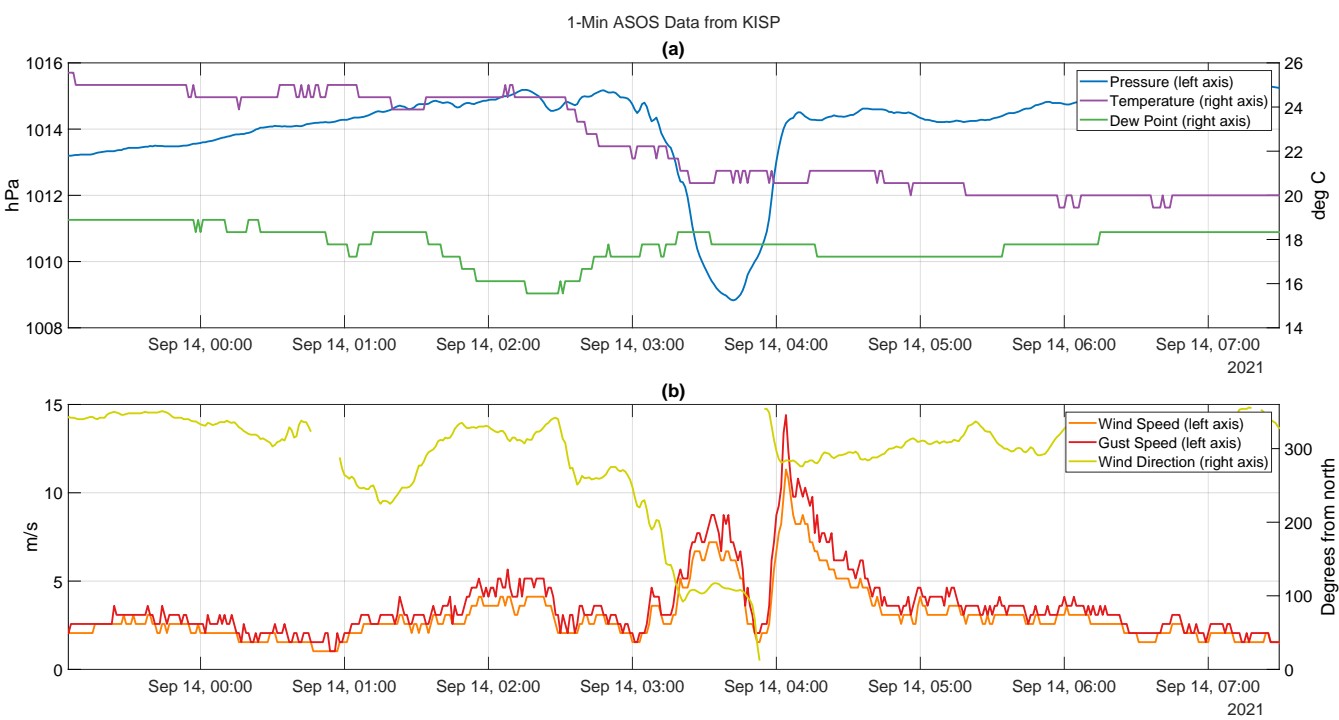

**Figure 17.** Time series of one-minute ASOS data from Islip, NY (KISP), on 14 September 2021. (a) Temperature (purple) dew point (red), and pressure (blue). (b) Wind speed (orange), wind gust speed (red), and wind direction in degrees clockwise from northerly (yellow). Wind direction is not plotted when it changes by more than $180°$ in consecutive observations (e.g., when crossing $0°$ or $360°$) or when the wind speed is below $1.5 \text{ m s}^{-1}$. All times are UTC.

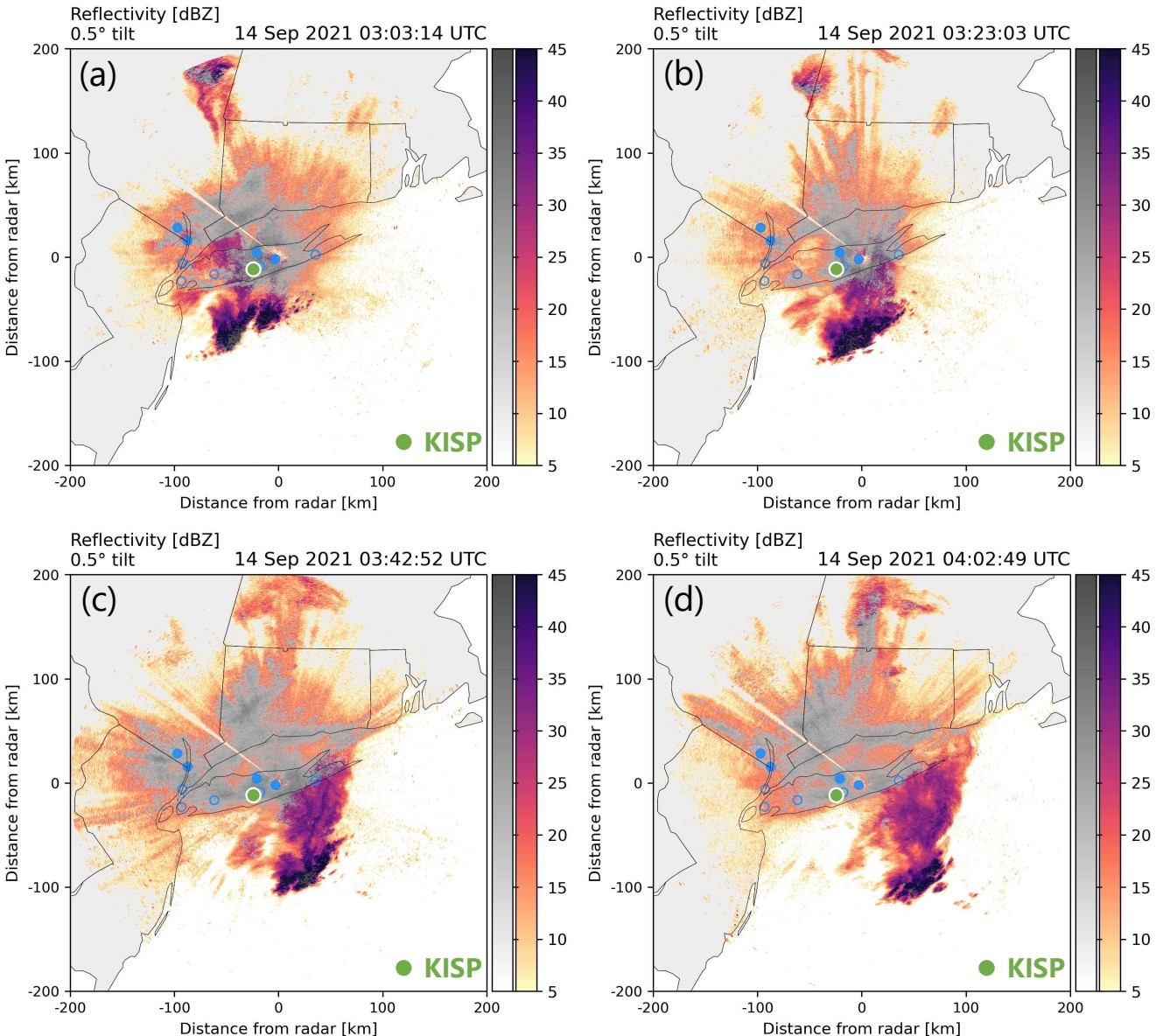

**Figure 18.** Maps of radar reflectivity at 0.5° tilt from the NWS WSR-88D radar in Upton, NY, on 14 September 2021. Reflectivity values in color show meteorological echo and those are shown in greyscale are likely non-meteorological echo such as insects and birds in this case (Tomkins et al., 2022). Filled blue circles indicate the locations of pressure sensors which captured the wave event described in Sect. 4.5, and unfilled blue circles indicate the locations of pressure sensors which did not capture the wave event. Filled green circle indicates the location of the Islip, NY, ASOS station (KISP). The sequence of images from (a) 0303 UTC, (b) 0323 UTC, (c) 0342 UTC, and (d) 0402 UTC shows the southeastward movement of region of convective cells > 40 dBZ from closer to further off the southern coast of Long Island. The wake low is inferred to be near the trailing edge of the weaker stratiform precipitation region behind (west of) the convective cells. The minimum pressure at KISP associated with the wake low occurred near the time of the scan shown in (c). An animated version of this figure, with Doppler velocity wave detection, is shown in Video Supplement Animation-Figure-S03.

|  | Outbound | | | Rebound | |
|---|---|---|---|---|---|
|  | (i) | (ii) | (iii) | (i) | (ii) |
| Event Start UTC | 1509 1/15 | 1532 1/15 | 1606 1/15 | 0356 1/16 | 0652 1/16 |
| Event End UTC | 1631 1/15 | 1619 1/15 | 1714 1/15 | 0823 1/16 | 0729 1/16 |
| Wave Period mm:ss | 01:19 | 01:19 | 01:05 | 11:25 | 06:37 |
| Mean Amplitude hPa | 0.30 | 0.13 | 0.07 | 2.43 | 0.36 |
| $N_{sensors}$ | 4 | 7 | 7 | 14 | 7 |
| Mean Cross-Correlation | 0.83 | 0.84 | 0.77 | 0.89 | 0.97 |
| Phase Speed m s$^{-1}$ | 326.6 | 237.5 | 178.9 | 292.1 | 290.4 |
| Phase Direction Degrees CW from N | 64.2 | 93.1 | 56.8 | 261.9 | 262.0 |
| RMSE s | 1.04 | 25.19 | 74.19 | 17.90 | 8.21 |
| NRMSE | 0.015 | 0.023 | 0.035 | 0.014 | 0.043 |

**Table 1.** Summary of 5 wave events associated with the Hunga Tonga-Hunga Ha'apai eruption Lamb waves in January 2022. Wave events were subset to those with mean cross-correlation above 0.75, modeled delay time RMSE below 90 s, and modeled delay time NRMSE below 0.1. The start times shown are the earliest among sensors which captured a given event, and end times shown are the latest among sensors which captured a given event. Center wave periods and amplitudes are averaged across the sensors which captured a given event. The first 3 events shown [Outbound (i), (ii), and (iii)] are events associated with the initial outbound set of Lamb waves, and the last 2 events shown [Rebound (i) and (ii)] are events associated with the initial rebound set of Lamb waves.