# Peer review of "Objective identification of pressure wave events from networks of 1-Hz, high-precision sensors"

_EGUsphere, 2023_

## Author Response (AR1)

Referee comments
Author response
Quoted text in revised manuscript

Allen et al. present a network of COTS pressure sensors time synced to a central server. They show how these networks can record wavelike oscillations in the gravity wave period range, and track them across individual elements. The tracking system is developed using wavelet and cross correlation methods. Several examples are shown, including the Hunga-Tonga event, gravity wave trains, and wake lows. The authors note that this method could increase the dataset of trackable gravity waves, with implications for winter storms. They note they have a forthcoming publication that will describe this in more detail.

Allen et al. present a fairly convincing case that these COTS sensors can record coherent wavelike phenomena from a variety of sources, including distant volcanic eruptions and gravity waves from diverse meteorological phenomena. They outline an event detection and association method, although it is not clear if this is happening in near real time or is via postprocessing. The manuscript is clearly in AMT's scope. The paper would benefit from some more background on the relevance of the observations conducted, how the sensor network could be replicated or expanded, previous instances of COTS pressure/seismic sensors, and clarity on the detection thresholds used. I recommend it for publication after the authors take the following comments into account.

We have clarified that the processing method shown in the manuscript is primarily intended for post-processing the data for research rather than near real time processing. We also tried to add further context to other low cost sensor networks and clarify our reasoning for choosing the detection thresholds shown in the paper, as discussed in the following comment responses.

MAJOR COMMENTS

1. It is unclear to me how this measurement technique will be used to either augment operational forecast models or conduct atmospheric studies. Even a methods-based paper needs to provide this information in order to motivate the reader to assess the technique in question. In the summary, the authors note that a forthcoming publication will detail the application of this measurement technique, but that is mentioned by afterthought. The introduction gives a good overview of gravity waves and what phenomena they impact, but I would appreciate more discussion of how this ties in with the network concept they have created. I encourage the authors to explicitly describe what measurement gaps they are trying to fill, and the impact such filling will have. I suggest inserting this discussion between the second to last and last paragraph in the introduction.

We have added the following paragraph discussing the measurement gap with examples of where that measurement gap is relevant:

"There is a scarcity of data for detecting and tracking pressure disturbances, including gravity waves, on the meso-β-scale or meso-γ-scale (ranging from 2 km to 200 km). Pressure disturbances on those scales may be relevant to phenomena such as snow bands (e.g., McMurdie et al., 2022), trade wind cumulus (e.g., Seifert and Heus, 2013), and bow echoes (e.g., Adams-Selin and Johnson, 2010), which are active areas of research. We developed a measurement and analysis technique which will allow for questions

regarding gravity waves on those scales to be addressed. To what degree and in what conditions information on gravity wave occurrence would be valuable in operational weather settings is yet to be determined."

We added the following sentence in the penultimate paragraph of the introduction:

"The methodology is intended to be used for post-processing in research applications, rather than for real time or near real time detection of wave events."

To illustrate the process by which we might expect gravity waves to modify cloud and precipitation processes, we contour relative humidity with respect to ice and with respect to liquid water as a function of temperature and water vapor mixing ratio in a standard atmosphere in Fig. R1. Gravity waves would lift up and drop down air parcels while holding the water vapor mixing ratio constant, as long as no phase changes occur. Lifted air parcels would experience cooling and thus an increase in relative humidity. Once the RH reaches 100% (dashed contours in Fig. R1), cloud would form. If the parcel then yields precipitation which falls out, then it is irreversibly changed by the lifting process (the subsequent descent due to the gravity wave would then occur at a lower water vapor mixing ratio).

*Figure R1. (a) Relative humidity with respect to ice (RH$_{ice}$) and with respect to liquid water (RH$_{water}$) as a function of temperature and water vapor mixing ratio for a standard atmosphere. Values are contoured at 20% intervals, starting at 20%, with the 100% contour dashed for both RH$_{ice}$ and RH$_{water}$. The temperature axis is inverted such that colder temperatures are at the top. (b) The temperature difference between the 100% RH$_{ice}$ and 100% RH$_{water}$ contours at each water vapor mixing ratio.*

[Figure]

We have added the following sentences to the first paragraph of the introduction:

"Within the troposphere, the vertical motions associated with gravity waves have been shown to influence cloud and precipitation processes."

We have also added new paragraphs to the introduction immediately after the first paragraph which explicitly describes how the vertical motions associated with gravity waves can impact cloud microphysics. Figure R1 has been added to the manuscript as Fig. 1. The new paragraphs say:

"To illustrate how the vertical motions associated with gravity waves could influence cloud microphysical properties, we use Fig. 1, which shows relative humidity with respect to ice (RH$_{ice}$) and with respect to liquid water (RH$_{water}$) as a function of temperature and water vapor mixing ratio for a standard atmosphere, as well as the temperature difference between 100% RH$_{ice}$ and 100% RH$_{water}$ for each water vapor mixing ratio. For temperatures below 0°C, RH$_{ice}$ > RH$_{water}$.

A lifted parcel would be cooled at a constant water vapor mixing ratio (move upward in Fig. 1) until it intersects the 100% RH$_{ice}$ contour wherein vapor deposition would reduce the water vapor (further lifting would move upward and to the left in Fig. 1). If the parcel cools sufficiently to also intersect the 100% RH$_{water}$ contour, supercooled water droplets would form in the parcel and riming would likely occur further depleting the available water vapor in the parcel. An up and down motion of an air parcel in a

gravity wave that only crosses the 100% $RH_{ice}$ contour yields ice mass changes that are reversible, i.e. the ice mass added by vapor deposition in the upward motion will be lost to sublimation in the downward motion.

In contrast, if the lifting of the parcel within a gravity wave starts at $RH_{ice} \geq 100\%$ and intersects the 100% $RH_{water}$ contour, the ice mass added by riming when ice particles collide with supercooled droplets is not reversible (i.e., there is no unriming process). For example, if an ice saturated parcel containing ice crystals (e.g., at -8.0°C and 2.96 g $kg^{-1}$ water vapor mixing ratio) is lifted and cooled to -9.7°C in a gravity wave (which requires ~190 m of lift assuming a 9°C $km^{-1}$ parcel lapse rate), it would become supersaturated with respect to water as well as with respect to ice. Water droplets would form in the parcel and the ice crystals could potentially become rimed. If the parcel remains ice saturated when it descends in the gravity wave, the additional ice mass from the riming on the ice particle would not be removed. Ice mass added by riming can only be removed by sublimation in conditions where the parcel is subsaturated with respect to ice."

2.  The low cost, COTS nature of this network is intriguing.   The paper would benefit from mentioning similar low cost network concepts, for example the Raspberry Shake seismmometer (see Anthony et al., DOI 10.1785/0220180251 and Lamb et al., DOI 10.3389/fcosc.2020.630967) and the RedVox infrasound cellphone app (Eaton et al, DOI 10.1016/j.apacoust.2022.109015, also see Garces et al, DOI: 10.3390/ signals3020014 and especially the reference list).  How does the presented architecture fit in with similar systems?  Could it be expanded to a broader network of pressure sensors, analogous to the Raspberry Shake seismic network>

We agree and have added mention of other low cost network concepts in the introduction and data sections, including context for the scale of our pressure sensor networks and the cost of each sensor. In the introduction, we added the following sentences:

"Similar low cost sensor networks have been used for detection of seismic (Anthony et al., 2018) and for detecting infrasound waves to monitor fan rotation speeds in nuclear reactor cooling towers (Eaton et al., 2022). The former network covered the area of Oklahoma, and the latter used networks covering the area of a single nuclear reactor. Our networks of pressure sensors are on the scale of a medium to large sized city or metropolitan area."

In the data section, we added the following sentences:

"Each pressure sensor costs roughly 50-75 US Dollars, subject to changes in the cost of Raspberry Pi Zero W units. For context to another low cost network concept, the Raspberry Shake 4D seismographs cost 'a few hundred dollars' per unit (Anthony et al., 2018)."

We don't think a deeper comparison of our sensors is appropriate for this manuscript. While there are many groups which make use of COTS sensors to research and citizen science applications the goals of each project are distinct from each other and our own. While we have a superficial similarity to the RaspberryShake network, they are very much ahead of us in terms of hardware and software maturity. Our sensors serve our simple needs, and we would require professional software development and a much more complex hardware backend to reach the level of sophistication of the Raspberry Shake. Reaching that level is an aspirational goal of ours and, conceptually, we could get there at some future point but too

much discussion on that issue would distract from the goal of our manuscript which is to establish a method of objective identification of wave events so that we can cite this method of pending papers that will show the fruits of our research.

3. Maybe not a "major comment" but one more reflecting my background in geophysical acoustics: The authors focus on gravity waves, but lower limit of the wave periods specified (1-3 minutes) include acoustic and acosutic gravity modes. The Hunga Tonga event is also mentioned in the examples. I suggest mentioning acoustic and acoustic/gravity waves in the abstract and the introduction because of this. Also, there are instances of very long period acoustic waves reported in the literature that may be meteorologically relevant. These include those from severe covective storms, especially those that generate hail (see Goerke and Woodward 1966 "Infrasonic observation of a severe weather system" and Elbing et al. (2019) DOI: 10.1121/1.5124486) Wind over mountains also generates powerful low frequency infrasound and may correlate with turbulence experienced by airplanes, see Bedard (1978) "Infrasound originating near mountainous regions in Colorado".

We agree that acoustic waves warrant mentioning. Coffer and Parker (2022) found that infrasound wave signals in tornadic and nontornadic supercells are quite similar, suggesting that there is no strong association between tornadogenesis and infrasound waves. Therefore it is unclear how much value infrasound analysis can add to convective storm research and forecasting.

We reworded the abstract to mention the shorter-period wave modes:

"Wave periods between 1 minute and 120 minutes were analyzed, a range which could capture acoustic, acoustic-gravity, and gravity wave modes."

We also now mention acoustic waves in the 2nd paragraph of the introduction:

"Acoustic and acoustic-gravity waves can also produce pressure signals at shorter time scales. Such waves would include infrasound waves, which can be associated with, e.g., convective storms and strong flow over mountains (Coffer and Parker, 2022; Bedard, 1978)."

4. There are statements made about the detection threshold of the sensors (e. g. lines 65-70), and what sort of periods and propagation speeds could be resolved. The statements made here would be easy to assess by generating synthetic data with certain background noise levels and features with various periods traveling at various velocities. These could be fed through the detection and association algorithm to determine which were captured and which were not. I recommend that this be done to more rigorously assess the detection ability of the networks presented here.

We have tested the detection of wave events with varying wave periods and amplitudes within a single pressure sensor by creating a time series of synthetic pressure data with normally distributed random noise centered on 0 with standard deviation equal to the noise floor (0.008 hPa). Sine waves with periods ranging from 2 min to 120 min and with amplitudes ranging from 0.01 hPa to 1 hPa were inserted into the time series. The sine waves each lasted for 2 hours, with a 12 minute ramp-up and ramp-down in amplitude at the start and end, and there were 2 hours in between each set of waves. The pressure time series, wavelet power, and event reconstruction for the synthetic event with an amplitude of +/- 1 hPa and wave period of 23 min are shown in Fig. R2.

Fig. R2 shows which events were detected and which were undetected using a K value of 10 (Eq. 4). Fifty-two out of the 105 wave events were detected, with no false positives. The weakest wave event had an amplitude of +/- 0.0464 hPa (or 0.0928 hPa from peak to trough). It does turn out that at lower K values, more of these wave events are detected, without many false positives. Even for K = 2, 86 out of the 105 wave events were detected, with only 1 false positive.

*Figure R2. (a) Part of the synthetic time series of pressure data (blue) with an extracted wave event (black). This synthetic wave event had a wave period of 2 min and amplitude of ±0.0464 hPa. (b) The wavelet power corresponding to the time series shown in (a). The dashed and solid contours indicate where the wavelet power is 5 times and 10 times the mean wavelet power for a given wave period, respectively. (c) The synthetic wave events which were detected (blue filled circles) and undetected (red X symbols) using our methodology with a K value of 10, as a function of their wave period and amplitude.*

[Figure]

Still, this exercise likely fails to capture the full extent of noise and interference of many signals present in the real pressure data. Across the 41 months of pressure data we have processed in Toronto and New York using the thresholds shown in the paper, 35 robust and trackable wave events were found using the thresholds stated in the manuscript, and we are confident that two of those were false positives, based on their short wave periods (roughly 1 minute), low amplitudes (below 0.2 hPa), and unusual phase velocities compared to the other 33 wave events (one was slower than any of the other 33, and the other had a strong westward component while the other 33 each had eastward components to their phase velocities). The pressure time series and extracted event waveforms for one of these cases are shown in Fig. R3. The weakest detected wave events with relatively short durations often have similar waveforms which can be erroneously paired together across sensors, resulting in cases like the two false positives discussed here. Lowering the K threshold would result in more of these erroneous pairings of events, and thus more false positives like the two discussed here. To be clear, events like the one shown in Fig. R3 are likely "real" at the particular location of each individual sensor, but the pairing across sensors is likely erroneous, so there is some degree of "interaction" between each of the thresholds we use (i.e., lowering our K threshold would probably need to be combined with increasing our cross-correlation or phase velocity error threshold).

We have added a sentence to the 3rd paragraph of Sect. 3.1 which says:

"This threshold can be adjusted, and different applications may warrant different values of K."

We have also added Fig. R2 to the manuscript as Fig. 6, and we added the following paragraph to the end of Sect. 3.1:

"We tested the method of detecting wave events in a single sensor using synthetic pressure data. The synthetic time series of pressure is created by an initial constant pressure value (which is randomly chosen from a normal distribution with mean 1000 hPa and standard deviation 2 hPa). We then added normally distributed random noise centered on 0 with standard deviation equal to the noise floor (0.008 hPa). Finally, we added 105 pre-defined wave events, which consist of sine waves of period ranging from 2 min to 120 min and maximum amplitude ranging from $\pm$0.01 hPa to $\pm$1 hPa. Each set of sine waves lasts for 2 hours, with a 12 minute ramp-up and ramp-down period at the start and end of those 2 hours in which the amplitude increases and decreases linearly, respectively. One of these synthetic wave events is shown in Fig. 6. Using the $\langle |W(b, a)| \rangle_b$ values shown in Fig. 5 and a K value of 10, 52 of the 105 synthetic wave events were detected, with no false positive event detections (Fig. 6c). The weakest detected synthetic wave event had an amplitude of $\pm$0.0464 hPa (or 0.0928 from peak to trough) and a wave period of 2 min (shown in Fig. 6a). Lower values of K do lead to more wave events being detected, with few false positives. The K = 2 used by Grivet-Talocia and Einaudi (1998) results in 86 of the 105 wave events being detected with only one false positive. However, this exercise likely fails to capture the full extent of noise and the interference of many signals present in real pressure data. Lower values of K can result in more weak pressure wave events being detected, which may be "real" at a single sensor location, but these may then erroneously paired with other weak pressure wave events at other sensor locations using the methods described in the proceeding sections. Therefore, we will use K = 10 to detect and track pressure wave events across the networks of sensors."

*Figure R3. Total pressure (blue, right axes) and extracted wave event (black, left axes) time series for a pressure wave event on 7 February 2020 in New York and Long Island. This event was estimated to have a phase speed of 18.9 m s⁻¹ at 219 degrees from north, with RMSE and NRMSE in that estimate of 13.9 s and 0.0044, respectively, but this is thought to be a false positive case.*

[Figure]

MINOR COMMENTS

Line 5:  Wave periods up to ~300 seconds are acoustic, or acoustic-gravity, modes.

We changed this sentence to include those wave modes. The sentence now reads:

"Wave periods between 1 minute and 120 minutes were analyzed, a range which could capture acoustic, acoustic-gravity, and gravity wave modes."

Line 25:

The signal generated by Hunga Tonga is exceptionally rare and has only been generated twice in my knowledge - Hunga Tonga and the Krakatoa eruption of 1883.  It may be worth mentioning that large bolide impacts or thermonuclear explosions can generate similar phenomena as well.  See Pierce and Posey (1971) "Theory of the excitation and propagation of Lamb's atmospheric edge mode from nuclear explosions".  The authors may have also meant gravity waves generated by large volcanic plumes, which are somewhat more common (though still quite rare, especially in eastern North America!).

We changed the beginning of this paragraph to:

"One way of distinguishing gravity waves from other wave phenomena such as Kelvin-Helmholtz waves is that gravity waves produce a surface pressure signal (Nappo, 2013, Sect. 8.2), given that the stable layer in which they occur is adjacent or nearly adjacent to the surface. That said, several

different phenomena can produce surface pressure disturbances on similar spatiotemporal scales to gravity waves, including but not limited to outflow boundary passages, convective wake lows (Johnson and Hamilton, 1988), release of conditional symmetric instability (Gray et al., 2011), and Lamb waves generated by, e.g., distant volcanic eruptions (Matoza et al., 2022), large bolide impacts (ReVelle, 2008), and thermonuclear explosions (Pierce and Posey, 1971)."

Lines 65-70 Is this detection threshold based on any simulations or analysis, or is it just a guess?

This value (0.08 hPa) was a guess derived from multiplying the noise floor (0.008 hPa) by 10. We have reworded the sentence to avoid giving an exact value which may be misleading. The sentence now reads:

"In order to reliably detect a wave signal, the amplitude likely needs to be substantially larger than the noise floor."

Line 120: Again, a study using synthetic data would be useful to better assess what K values to use.

See above discussion under Major Comment 4 about sensitivity of our wave detection method, our choice of K = 10, and changes to this paragraph.

Lines 145-150: This 2 hour threshold seems arbitrary. Doesn't it depend on the separation distance between the pair of furthest-apart sensors in a given array, divided by the typical propagation speeds of the wave features of interest?

The largest distance between sensors for the New York/Long Island network is 136 km (sensor 027 to sensor 012, Fig. 1c). If a wave feature propagated parallel to the distance vector between those sites, an event with phase speed 18.9 m/s would be the slowest that could be paired over those 2 sensors with this 2 hour threshold. However, waves propagating at an angle relative to the distance vector between those sensors could be slower and still be paired across those 2 sensors. Also, the processing technique should allow for sensors in between the 2 most distant sensors to "bridge the gap" (e.g., sensor 027 to sensor 020 to sensor 011 to sensor 012, Fig. 1c), and it would be very unusual, if not impossible, for a pressure wave event to occur at sensors 027 and 012 without also occurring at the sensors in between. In any case, we need a wave event to be detected by 4 sensors to be considered trackable. Finally, the time between the end of an event for one sensor and the start of the event for the other sensor (which is what the 2 hour threshold refers to) would be less than the time between actual wave front passages for the two sensors. Therefore, 18.9 m/s is a conservative estimate of the minimum phase speed for an event that could be tracked from sensor 027 to sensor 012 using our methods.

The same calculation for the Toronto and Raleigh networks gives minimum phase speeds of 4.3 m/s and 3.1 m/s, respectively, which are below the expected characteristic phase speeds for gravity waves. For the combined sensor network used to track the Lamb waves produced by the Hunga Tonga eruption, the same calculation gives a minimum phase speed of 123.8 m/s.

We added the following sentences to that paragraph:

"This 2 hour threshold is subjective, and it affects the range of speeds of wave events which can be detected. The threshold can be altered depending on the distances between pressure sensors and the

desired application. For example, the largest distance between any 2 sensors within any of the 3 networks is that between sensors 012 and 027 (136 km apart, Fig. 1). A wave feature propagating at 18.9 m s$^{-1}$ would take 2 hours to propagate that distance. However, wave features propagating at an angle could be slower and propagate over both sensors within 2 hours, and as long as pressure sensors in between those two most distant sensors capture the event, the following processing technique will allow those sensors to "bridge the gap". Thus, 18.9 m s$^{-1}$ is a conservative estimate of the minimum phase speed required for this methodology to track a wave event."

Line 201 - What kind of 'testing'?

"Testing" is perhaps the wrong word here. We processed 41 months of data for both the Toronto and New York pressure sensor networks, and considered the resulting distributions of RMSE and NRMSE for each of the wave events captured by at least 4 sensors. Figure R4 shows those distributions along with the thresholds we chose. There were a large number of events which clearly had too much error to consider trackable (on the order of tens of minutes to hours in RMSE). Conversely, there was a very dense cluster of events with NRMSE and RMSE near 0 (lower-left in Fig. R4b). This makes sense; if the lag times between sensors are consistent with each other (e.g., the lag times from sensor A to sensor B, sensor B to sensor C, and sensor C to sensor A sum to near 0), the error in the least-squares calculation of the slowness vector should be small. Therefore, we were most confident in the phase velocity calculation for that cluster of points. Selecting the particular thresholds of RMSE and NRMSE (90 s and 0.1, respectively) from there was ultimately a subjective choice (e.g., RMSE thresholds of either 50 or 150 would have also been reasonable).

We changed this sentence to now read:

"After processing multiple years of data from the Toronto and New York pressure sensor networks and analyzing the resulting RMSE and NRMSE distributions, it was found that a maximum RMSE threshold of 90 s and a maximum NRMSE threshold of 0.1 are reasonable to consider a wave event trackable."

*Figure R4. NRMSE against RMSE for wave events detected by at least 4 sensors in the Toronto and New York pressure sensor networks. In (a), all wave events are shown. In (b), only wave events with RMSE below 500 s are shown. The RMSE and NRMSE thresholds of 90 s and 0.1, respectively, to consider a wave event trackable are indicated by the purple outline.*

[Figure]

Line 215: The phase speed is not precisely the speed of sound - better to say "the phase speed is known". The waves from Hunga Tonga were not really sound waves in the normal sense of the word.

We changed this sentence to:

"The Lamb waves caused by the Hunga Tonga-Hunga Ha'apai volcanic eruption in January 2022 is a case where the origin of the waves is known and the phase speed is known as a function of air temperature (Amores et al., 2022)."

Line 223 - The ash plume technically reached the mesosphere

Line 225 - It went a lot higher than 30 km (55 km, see Kloss et al., DOI: 10.1029/2022GL099394)

Kloss et al. do not show this directly; they cite the satellite analyses of Carr et al. (2022) and Proud et al. (2022). Therefore, we changed these sentences to:

"On 14-15 January, 2022, large eruptions occurred at the Hunga Tonga volcano in the south Pacific Ocean which produced ash plumes reaching the mesosphere and a series of shock waves. A particularly violent, submarine eruption occurred at around 0400 UTC 15 January (Global Volcanism Program, 2022). Satellite data suggest that the ash plume associated with this eruption reached as high as 57 km a.s.l. (Carr et al., 2022; Proud et al., 2022) and contained roughly 400 million kg of sulfur dioxide."

Line 233 - The signals were not 'shock waves' when they reached the sensors. The term 'shock wave' denotes a specific nonlinear process that was not at work here.

Line 244 - Lamb waves, not Lamb shock waves

We have revised this section and relevant table and figure captions to avoid using the term "shock waves" with regard to the pressure waves we observed, and now "Lamb waves" appears consistently, except in the first sentence of this subsection, where we mention that shock waves were produced near the eruption site.

Figures 1 and 4 are never referred to in the text.

These figures (renumbered to figures 2 and 5) were referred to on lines 48, 122, and 232 in the original manuscript, and now appear on lines 93, 169, 197, 215, and 314 in the revised manuscript.

GENERAL COMMENTS

The manuscript "Objective identification of pressure wave events from networks of 1-Hz, high-precision sensors" by L.R. Allen et al. describes the study of pressure wave events (gravity wave like) with small arrays of low-cost pressure sensors. The instrumentation, wavelet and array processing methodology, data processing as well as a number of five exemplary event cases are provided and serve as a kind of proof of concept for the given approach. The study focuses on a link of the gravity wave signatures to meteorological phenomena like clouds and precipitation and an outlook is given on a further systematic climatological investigation in that context.

The study shows good quality in the scientific approach and presentation of methods and results. It provides a clear step-by-step description of the measurement and processing techniques, making it suitable for the AMT journal. Anyhow, neither the pressure measurements of acoustic, infrasonic, acoustic-gravity or buoyancy waves, nor the array processing of distributed, low-cost sensors, the cross correlation as well as travel time difference calculation for velocity estimation is a novel, innovative or substantially new approach. Therefore the general significance of the concept and results to identify high-amplitude events is comparably low. Acceptance of the study for publication therefore depends on certain clarifications and improvements of the manuscript, as specifically commented below, provided by RC1 and probably supported by additional reviewer comments.

 We have tried to clarify the data gap we are trying to fill with our pressure sensors and add context to other low cost sensor networks. We have also added more discussion about the potential research applications of our data and methods.

SPECIFIC COMMENTS

- Introduction and References Sections: the manuscript would benefit from some improved references to strengthen the respective contexts of gravity waves, volcanoes, meteorology and instrumentation. Therefore, in lines 12-13 a comprehensive peer-reviewed review paper on gravity waves and their effects should be added, I recommend Fritts and Alexander 2003 (https://doi.org/10.1029/2001RG000106) for that. The different phenomena in lines 24/25 that are also the basis for the examples later on, should be referenced from literature. I recommend papers connecting gravity wave (like) observations of volcanic and meteorological examples like Vergoz et al, 2022 (https://doi.org/10.1016/j.epsl.2022.117639) and Coleman and Knupp, 2009 (https://doi.org/10.1175/2009WAF2222248.1) or similar. Further phenomena as mentioned  by RC1 are also worthwhile. I also second the RC1 opinion to mention other distributed instrumentation approaches like Raspberry or RedVox. Furthermore the infrasound measurements of the USarray especially of gravity waves should be mentioned, see C. de Groot Hedlin et al, 2014 (https://doi.org/10.1016/j.epsl.2013.06.042), around lines 32 to 39.

We agree that the Fritts and Alexander (2003) paper is a useful citation for introducing gravity waves and their possible source mechanisms; however, they focus on gravity waves in the middle atmosphere, whereas our focus is on gravity waves near the surface. We have added a sentence in the first paragraph of the introduction addressing possible gravity wave sources:

"The initial perturbations which generate gravity waves can have several sources, including but not limited to forced flow over topography, deep convection, shear instability, adjustment of unbalanced flow, and nonlinear interaction between waves (Fritts and Alexander, 2003)."

We have added references for the different phenomena in lines 24/25 as discussed in our response to RC1:

"That said, several different phenomena can produce surface pressure disturbances on similar spatiotemporal scales to gravity waves, including but not limited to outflow boundary passages, convective wake lows (Johnson and Hamilton, 1988), release of conditional symmetric instability (Gray et al., 2011), and Lamb waves generated by, e.g., distant volcanic eruptions (Matoza et al., 2022), large bolide impacts (ReVelle, 2008), and thermonuclear explosions (Pierce and Posey, 1971)."

We have also added sentences referencing the Raspberry Shake sensor network and RedVox in the introduction as discussed in our response to RC1:

"Similar low cost sensor networks have been used for detection of seismic (Anthony et al., 2018) and for detecting infrasound waves to monitor fan rotation speeds in nuclear reactor cooling towers (Eaton et al., 2022). The former network covered the area of a U.S. state, and the latter used networks covering the area of a single nuclear reactor. Our networks of pressure sensors are on the scale of a medium to large sized city or metropolitan area."

We also added sentences in the Data section comparing the cost of our pressure sensors to those in the Raspberry Shake sensor network:

"Each pressure sensor costs roughly 50-75 US Dollars, subject to changes in the cost of Raspberry Pi Zero W units. For context to another low cost network concept, the Raspberry Shake 4D seismographs cost "a few hundred dollars" per unit (Anthony et al., 2018)."

Finally, we have added sentences about the study of USArray data by de Groot Hedlin et al.:

"de Groot-Hedlin et al. (2014) used 337 barometers deployed with the USArray Transportable Array, recording at 1 Hz (i.e., every 1 sec) frequency, to detect and track high amplitude (roughly 3 hPa peak to trough) pressure waves associated with convective storms in the southern United States. The USArray Transportable Array barometers were spaced roughly 70 km apart, which might also preclude tracking of localized disturbances."

- Data and Methods Sections: Data and methodology are clearly described with only some small questions open: e.g. why is an 2 hour time window chosen in line 147 and what is the specific, measurable impact of using four sensors in line 203 and not three, or let's say five. More generally it would be good, as also stated by RC1 to provide a sensitivity analysis on the impact of the described threshold, signal-to-noise ratio, wavelet and error limits chosen, and so on.

As discussed in our response to RC1, the 2 hour threshold on line 147 is somewhat subjective, but it does result in a lower limit on the phase speed of trackable waves through the network, with some caveats regarding the angle of wave propagation and the possibility of tracking waves through sensors in between the two that are farthest apart. We added the following sentences to that paragraph:

"This 2 hour threshold is subjective, and it affects the range of speeds of wave events which can be detected. The threshold can be altered depending on the distances between pressure sensors and the desired application. For example, the largest distance between any 2 sensors within any of the 3 networks is that between sensors 012 and 027 (136 km apart, Fig. 1). A wave feature propagating at 18.9 m s$^{-1}$ would take 2 hours to propagate that distance. However, wave features propagating at an angle could be slower and propagate over both sensors within 2 hours, and as long as pressure sensors in between those two most distant sensors capture the event, the following processing technique will allow those sensors to "bridge the gap". Thus, 18.9 m s$^{-1}$ is a conservative estimate of the minimum phase speed required for this methodology to track a wave event."

As mentioned in the final paragraph of the methods section, we discard events detected by only 3 sensors because the phase velocity calculation is less constrained, and could have smaller error values by chance. While this could still occur for an event detected by only 4 sensors, it appears to be much less likely. Using our thresholds of maximum RMSE of 90 s and maximum NRMSE of 0.1, there were 288 wave events meeting those criteria that were detected by at least 3 sensors in New York or Toronto between January 2020 and May 2023. Increasing the required number of sensors to 4 reduces that to 35 wave events, and increasing the required number of sensors to 5 reduces that to 14 wave events. There were several time periods when only 4 sensors were active in one or both of those networks, so we tried to avoid limiting the potential analysis times to only those when 5 sensors were active in a network.

In our response to RC1 (under major comment 4 and multiple minor comments) we show some results of sensitivity analysis on our wavelet power threshold (K = 10) and discussion on our choice of thresholds in RMSE, NRMSE, and the 2 hour maximum time separation between events in 2 sensors for those events to be paired together. We added a paragraph on that sensitivity analysis to the manuscript at the end of Sect. 3.1:

"We tested the method of detecting wave events in a single sensor using synthetic pressure data. The synthetic time series of pressure is created by an initial constant pressure value (which is randomly chosen from a normal distribution with mean 1000 hPa and standard deviation 2 hPa). We then added normally distributed random noise centered on 0 with standard deviation equal to the noise floor (0.008 hPa). Finally, we added 105 pre-defined wave events, which consist of sine waves of period ranging from 2 min to 120 min and maximum amplitude ranging from ±0.01 hPa to ±1 hPa. Each set of sine waves lasts for 2 hours, with a 12 minute ramp-up and ramp-down period at the start and end of those 2 hours in which the amplitude increases and decreases linearly, respectively. One of these synthetic wave events is shown in Fig. 6. Using the ⟨|W (b, a)|⟩$_b$ values shown in Fig. 5 and a K value of 10, 52 of the 105 synthetic wave events were detected, with no false positive event detections (Fig. 6c). The weakest detected synthetic wave event had an amplitude of ±0.0464 hPa (or 0.0928 from peak to trough) and a wave period of 2 min

(shown in Fig. 6a). Lower values of K do lead to more wave events being detected, with few false positives. The K = 2 used by Grivet-Talocia and Einaudi (1998) results in 86 of the 105 wave events being detected with only one false positive. However, this exercise likely fails to capture the full extent of noise and the interference of many signals present in real pressure data. Lower values of K can result in more weak pressure wave events being detected, which may be "real" at a single sensor location, but these may then erroneously paired with other weak pressure wave events at other sensor locations using the methods described in the proceeding sections. Therefore, we will use K = 10 to detect and track pressure wave events across the networks of sensors."

- Examples Section: to be even more specific, the question should be discussed if the given examples are only the raisins picked or if they represent typical events. This might shed light on the accuracy of detecting notable gravity wave events, on the true/false positive/negative rate of detections of events, and on the reliability of the approach in general. A receiver operation characteristic (ROC) analysis is a possible approach to do so. This might be a topic for open discussion during the review process, and not an obligation to be done, since the manuscripts summary already briefly discusses that there are several prerequisites and issues that are necessary and might allow or prohibit a gravity wave detection at the given surface instrumentation. That discussion should be enhanced taking into account influence factors for detection or non-detection of events and their significance. I would generally expect a more elaborated discussion section within the manuscript, taking into account the given and upcoming open discussion remarks.

We added a few sentences in the first paragraph of the examples section to clarify which examples were typical events and which were atypical events:

"These events were each manually chosen after roughly 40 months of pressure data were processed. The gravity wave train on 25 February 2022 and cold front example on 15 November 2020 are not unusual; several other gravity wave trains and pressure jumps due to cold front passages were found. The other three example cases are atypical events, however. It is extremely rare to detect a pressure signal due to a volcanic eruption several thousand km away. While other outflow boundaries were found to produce detectable pressure waves, the case on 4 February 2022 was unusual in terms of the time of year when it occurred (most other pressure waves associated with outflow boundary passages occurred during the warm season) and the radar signature. The wake low detected on 14 September 2021 was the only wake low event detected by our pressure sensor network."

As mentioned in our response to RC1, there were 35 total wave events meeting our criteria across 41 months of processed pressure data in Toronto and New York. We are confident that 2 of those were false positive detections. We show results of a sensitivity analysis using synthetic data in our response to RC1, which might suggest lower thresholds could be used, but there are complicating factors in the real data which are difficult to replicate in synthetic wave events.

We elaborated further on the strengths and weaknesses of our processing methods, including factors influencing detection or non-detection of wave events, in the Discussion and Summary section. The new sentences are:

"The methods shown are intended mainly for post-processing of pressure data for research applications, and not for real time, operational use. A benefit to this method is that it can be fully automated to detect wave events across many months of data, and we have made the processing code openly available (Allen and Miller, 2023)." …

"We use a rather strict criteria for detecting a wave event in a given sensor; the peak wavelet power must exceed 10 times the mean value for a given wave period (i.e., K = 10; eq. 4). Grivet-Talocia and Einaudi (1998) also used wavelet analysis to detect pressure waves with a scale dependent threshold; their K value was only 2. Therefore, many wave events detected using our criteria will be relatively high amplitude (on the order of 1 hPa or more), including most examples shown in Sect. 4. Depending on the desired application, this threshold and the other thresholds used in the wave detection can be adjusted.

Environmental factors can influence whether or not a given gravity wave is detected by our surface pressure sensors." … "Our method may not always properly track waves which are modified by local conditions (which may alter their amplitude, frequency, and/or phase velocity) as they propagate across the sensor network."

TECHNICAL COMMENTS:

the language as well as representation of formulae and figures is fine for me, I'm also in favor of the available data (and statement) and additional visual material. I have only minor technical remarks:

- line 215: doubling of "where"

Thank you for pointing this out; it has been fixed.

- line 224/225: the plume reached around 57 km and thus the mesosphere. The above given reference Vergoz et al could also be used around line 227 and following to reference observations of the lamb wave. According to Vergoz et al, it was detected during even five circles around the globe.

We have corrected the sentences on the height of the plume:

"On 14-15 January, 2022, large eruptions occurred at the Hunga Tonga volcano in the south Pacific Ocean which produced ash plumes reaching the mesosphere and a series of shock waves. A particularly violent, submarine eruption occurred at around 0400 UTC 15 January (Global Volcanism Program, 2022). Satellite data suggest that the ash plume associated with this eruption reached as high as 57 km a.s.l. (Carr et al., 2022; Proud et al., 2022) and contained roughly 400 million kg of sulfur dioxide."

Vergoz et al. (2022) say "Four full Lamb wave circulations around the globe can be confirmed" in the first paragraph Section 4.2. We reworded this sentence to say:

"The pressure signal associated with the Lamb wave was observed to circle the Earth several times with estimated phase speeds exceeding 100 m s$^{-1}$ (Adam, 2022; Burt, 2022; Vergoz et al., 2022)."

- line 242: "phase speed of 261.9", according to table 1 this is the azimuth and the phase speed is 292.1

Thank you, this has been corrected.

- line 267: name the number of detecting sensors for this case (five?)

We have reworded to:

"Between 1730 and 1900 UTC on 4 February 2022, an event with amplitude of roughly 1.8 hPa was detected by five pressure sensors in the New York City metro area and Long Island."

- Caption of figure 6: "Hunga Ha'apai" instead of " Hunga Ta'apai"

Thank you for pointing this out; it has been fixed.

This manuscript presents a method to extract signals from small networks of low-cost, high-precision pressure sensors. This topic fits the scope of the journal in terms of data processing techniques that can be applied to atmospheric measurements. The manuscript has fairly clear motivations, is logically organized, written well, and includes appropriate figures and supplementary information. The methods are clearly stated and enough detail is provided so that these may be replicated in other studies. The examples are concise descriptions of an assortment of phenomena that drive the observed pressure signals, and sufficiently highlight the method's ability to detect the associated signal for these different cases.

There is only one additional minor suggestion. The first sentence of the abstract could be split into two with one stating the causes of pressure waves, and the second on why they are important, such as their impact on clouds and precipitation. Similarly in the introduction, it might make more sense to state what they are, how they are generated, and how to distinguish them before discussing their potential impact on cloud and precipitation processes.

We added the following sentence to the abstract to try to address the impact of the waves on clouds and precipitation:

"The vertical motions associated with these waves can modify the temperature and relative humidity of air parcels and thus yield potentially irreversible changes to the cloud and precipitation content of those parcels."

We added the following sentences to the first paragraph of the introduction to address the potential origins of gravity waves and elaborate on how they can modify clouds and precipitation, as also discussed in our response to RC1:

"The initial perturbations which generate gravity waves can have several sources, including but not limited to forced flow over topography, deep convection, shear instability, adjustment of unbalanced flow, and nonlinear interaction between waves (Fritts and Alexander, 2003). Within the troposphere, the vertical motions associated with gravity waves have been shown to influence cloud and precipitation processes."

We also added a new paragraph following the first paragraph in the introduction which describes how the vertical motions associated with gravity waves influence cloud microphysics, and a new Fig. 1 to go with that paragraph. The new paragraph says:

"To illustrate how the vertical motions associated with gravity waves could influence cloud microphysical properties, we use Fig. 1, which shows relative humidity with respect to ice ($RH_{ice}$) and with respect to liquid water ($RH_{water}$) as a function of temperature and water vapor mixing ratio for a standard atmosphere, as well as the temperature difference between 100% $RH_{ice}$ and 100% $RH_{water}$ for each water vapor mixing ratio. For temperatures below 0°C, $RH_{ice} > RH_{water}$.

A lifted parcel would be cooled at a constant water vapor mixing ratio (move upward in Fig. 1) until it intersects the 100% $RH_{ice}$ contour wherein vapor deposition would reduce the water vapor (further lifting would move upward and to the left in Fig. 1). If the parcel cools sufficiently to also intersect the 100% $RH_{water}$ contour, supercooled water droplets would form in the parcel and riming would likely occur

further depleting the available water vapor in the parcel. An up and down motion of an air parcel in a gravity wave that only crosses the 100% $RH_{ice}$ contour yields ice mass changes that are reversible, i.e. the ice mass added by vapor deposition in the upward motion will be lost to sublimation in the downward motion.

In contrast, if the lifting of the parcel within a gravity wave starts at $RH_{ice} \geq 100\%$ and intersects the 100% $RH_{water}$ contour, the ice mass added by riming when ice particles collide with supercooled droplets is not reversible (i.e., there is no unriming process). For example, if an ice saturated parcel containing ice crystals (e.g., at -8.0°C and 2.96 g kg$^{-1}$ water vapor mixing ratio) is lifted and cooled to -9.7°C in a gravity wave (which requires ~190 m of lift assuming a 9°C km$^{-1}$ parcel lapse rate), it would become supersaturated with respect to water as well as with respect to ice. Water droplets would form in the parcel and the ice crystals could potentially become rimed. If the parcel remains ice saturated when it descends in the gravity wave, the additional ice mass from the riming on the ice particle would not be removed. Ice mass added by riming can only be removed by sublimation in conditions where the parcel is subsaturated with respect to ice."
* * *
This manuscript presents a method for extracting signals from a small network of low-cost, high-precision pressure sensors. The topic of the manuscript is clear and logically organized. The manuscript makes a compelling case that these COTS sensors can record coherent wave-like phenomena from a variety of sources, including distant volcanic eruptions and gravity waves from a variety of meteorological phenomena.

It is recommended that gravity waves and their effects on clouds and precipitation be described in more detail again in the first part, i.e., explaining what gravity waves are, how they are generated, and what kinds there are.

We have changed the first paragraph of the introduction to elaborate more on the possible generation mechanisms for gravity waves:

"The initial perturbations which generate gravity waves can have several sources, including but not limited to forced flow over topography, deep convection, shear instability, adjustment of unbalanced flow, and nonlinear interaction between waves (Fritts and Alexander, 2003). Within the troposphere, the vertical motions associated with gravity waves have been shown to influence cloud and precipitation processes."

We have also added Fig. 1 and a new paragraph following the first paragraph of the introduction which describes how the vertical motions associated with gravity waves can influence cloud microphysics. The new paragraph says:

"To illustrate how the vertical motions associated with gravity waves could influence cloud microphysical properties, we use Fig. 1, which shows relative humidity with respect to ice ($RH_{ice}$) and with respect to liquid water ($RH_{water}$) as a function of temperature and water vapor mixing ratio for a standard atmosphere, as well as the temperature difference between 100% $RH_{ice}$ and 100% $RH_{water}$ for each water vapor mixing ratio. For temperatures below 0°C, $RH_{ice} > RH_{water}$.

A lifted parcel would be cooled at a constant water vapor mixing ratio (move upward in Fig. 1) until it intersects the 100% $RH_{ice}$ contour wherein vapor deposition would reduce the water vapor (further lifting would move upward and to the left in Fig. 1). If the parcel cools sufficiently to also intersect the 100% $RH_{water}$ contour, supercooled water droplets would form in the parcel and riming would likely occur further depleting the available water vapor in the parcel. An up and down motion of an air parcel in a gravity wave that only crosses the 100% $RH_{ice}$ contour yields ice mass changes that are reversible, i.e. the ice mass added by vapor deposition in the upward motion will be lost to sublimation in the downward motion.

In contrast, if the lifting of the parcel within a gravity wave starts at $RH_{ice} \geq 100\%$ and intersects the 100% $RH_{water}$ contour, the ice mass added by riming when ice particles collide with supercooled droplets is not reversible (i.e., there is no unriming process). For example, if an ice saturated parcel containing ice crystals (e.g., at -8.0°C and 2.96 g kg$^{-1}$ water vapor mixing ratio) is lifted and cooled to -9.7°C in a gravity wave (which requires ~190 m of lift assuming a 9°C km$^{-1}$ parcel lapse rate), it would become supersaturated with respect to water as well as with respect to ice. Water droplets would form in the parcel and the ice crystals could potentially become rimed. If the parcel remains ice saturated when it descends in the gravity wave, the additional ice mass from the riming on the ice particle would not be

removed. Ice mass added by riming can only be removed by sublimation in conditions where the parcel is subsaturated with respect to ice."

The manuscript may need to describe how this measurement technique will be used to enhance operational forecasting models or for atmospheric research.

We have clarified that we intend for the methods in the paper to be used for post-processing in research applications in the fifth paragraph of the introduction. We also added the fourth paragraph of the introduction, which elaborates on the measurement gap we are trying to fill and the potential applications of the technique:

"There is a scarcity of data for detecting and tracking pressure disturbances, including gravity waves, on the meso-$\beta$-scale or meso-$\gamma$-scale (ranging from 2 km to 200 km). Pressure disturbances on those scales may be relevant to phenomena such as snow bands (e.g., McMurdie et al., 2022), trade wind cumulus (e.g., Seifert and Heus, 2013), and bow echoes (e.g., Adams-Selin and Johnson, 2010), which are active areas of research. We developed a measurement and analysis technique which will allow for questions regarding gravity waves on those scales to be addressed. To what degree and in what conditions information on gravity wave occurrence would be valuable in operational weather settings is yet to be determined."